# Reconstructing unseen transmission events to infer dengue dynamics from viral sequences

Henrik Salje[1,2,3,4 ✉], Amy Wesolowski [4], Tyler S. Brown[5], Mathew V. Kiang [5,6], Irina Maljkovic Berry [7], Noemie Lefrancq [1,2], Stefan Fernandez[8], Richard G. Jarman [7], Kriangsak Ruchusatsawat[9], Sopon Iamsirithaworn[10], Warunee P. Vandepitte[11], Piyarat Suntarattiwong[11], Jonathan M. Read[12], Chonticha Klungthong[8], Butsaya Thaisomboonsuk[8], Kenth Engø-Monsen [13], Caroline Buckee [5,15], Simon Cauchemez [2,15] & Derek A. T. Cummings [3,14,15]

For most pathogens, transmission is driven by interactions between the behaviours of infectious individuals, the behaviours of the wider population, the local environment, and immunity. Phylogeographic approaches are currently unable to disentangle the relative effects of these competing factors. We develop a spatiotemporally structured phylogenetic framework that addresses these limitations by considering individual transmission events, reconstructed across spatial scales. We apply it to geocoded dengue virus sequences from Thailand (N = 726 over 18 years). We find infected individuals spend 96% of their time in their home community compared to 76% for the susceptible population (mainly children) and 42% for adults. Dynamic pockets of local immunity make transmission more likely in places with high heterotypic immunity and less likely where high homotypic immunity exists. Age-dependent mixing of individuals and vector distributions are not important in determining spread. This approach provides previously unknown insights into one of the most complex disease systems known and will be applicable to other pathogens.

[1] Department of Genetics, University of Cambridge, Cambridge, UK. [2] Mathematical Modelling of Infectious Diseases Unit, Institut Pasteur, CNRS, UMR 2000 Paris, France. [3] Department of Biology, University of Florida, Gainesville, FL, USA. [4] Department of Epidemiology, Johns Hopkins Bloomberg School of Public Health, Baltimore, MD, USA. [5] Department of Epidemiology, Harvard T.H. Chan School of Public Health, Boston, MA, USA. [6] Department of Epidemiology and Population Health, Stanford University School of Medicine, Stanford, CA, USA. [7] Viral Diseases Branch, Walter Reed Army Institute of Research, Silver Spring, MD, USA. [8] Department of Virology, Armed Forces Research Institute of Medical Sciences, Bangkok, Thailand. [9] National Institute of Health, Department of Medical Sciences, Ministry of Public Health, Nonthaburi, Thailand. [10] Department of Disease Control, Ministry of Public Health, Nonthaburi, Thailand. [11] Queen Sirikit National Institute of Child Health, Bangkok, Thailand. [12] Lancaster Medical School, Lancaster University, Lancaster, UK. [13] Telenor Research, N-1360 Fornebu, Norway. [14] Emerging Pathogens Institute, University of Florida, Gainesville, FL, USA. [15]These authors contributed equally: Caroline Buckee, Simon Cauchemez, Derek A. T. Cummings. ✉email: hs743@cam.ac.uk

As with other endemic pathogens, widespread, sustained co-circulation of dengue viruses (DENV), effectively masks the dynamics of individual lineages[1–3]. The co-occurrence of unrelated transmission chains means we still only have a limited understanding of how DENV spreads, including the role for human mobility of both infected individuals and the surrounding susceptible population, age-specific mixing and local heterogeneities in serotype-specific population immunity and mosquito density. These mechanistic knowledge gaps help explain our failures to control a pathogen that continues to cause 50 million annual symptomatic infections globally[4]. The use of pathogen sequences has the potential to help. However, existing phylogeographical approaches can only provide limited mechanistic insight into drivers of spread as they focus on rates of flow between locations present in a phylogeny, based on assumptions of mass action and using traits attached to the observed sequences (i.e., cases) only[5,6]. In addition, fewer than 1% of dengue infections will currently be sequenced from any one country[4]. Critically, existing phylogeographic approaches do not consider that viral flow is made up of sequential transmission events with each event arising from a complex interplay of individual-, population-, environmental- and viral-level factors. Further, the bulk of available sequences typically come from a few locations with most locations providing no data. Many existing phylogeographic approaches will infer a viral flow between observed locations without consideration that transmission events that link two observed sequences will be unobserved and often in unsampled locations.

Here we develop an inference framework that fills this knowledge gap by explicitly considering individual transmission events. By using the generation time distribution for dengue (Supplementary Fig. 1), we derive estimates of the number of generations that separate each pair of sequences in a time-resolved phylogeny and consider viral mobility for a single-transmission generation. This shift of focus to single-transmission generations, rather than overall viral flow, allows us to develop detailed mechanistic models of how viruses are moving at a tractable and interpretable scale. For example, we separately

model population movement for infected individuals, the susceptible population (mainly children) as compared to adults, allowing for transmission to occur in the infector's community, in the infectee's community or in a tertiary location. We assume the local scale of movement of the *Aedes* mosquito means only human mobility can drive spread between locations[7]. We allow for the disabling symptoms from dengue to result in reduced mobility in cases compared to the susceptible population[8]; that transmission may occur in an age-structured manner[3], and that the spatial heterogeneity in vector distributions and the dynamic nature of local serotype-specific population immunity may affect where successful transmissions occur[9]. Using the transmission probabilities for a single generation, we probabilistically integrate over all possible pathways for the total number of transmission generations that link the observed locations for each pair of sequences, thereby capturing movement in unsampled locations. In parallel, we incorporate the probability of sequencing (i.e., observing) an infection at each space–time unit, thereby explicitly incorporating space–time biases in sampling. We fit our models in a maximum likelihood framework that incorporates uncertainty from the evolutionary processes, including topical uncertainty in the phylogenetic trees, uncertainty in the generation time distribution and sampling uncertainty using a bootstrap approach.

We apply our framework to dengue in Thailand, a country that has suffered from self-sustained dengue circulation for decades[3,10]. We use 726 sequences obtained from seven different provinces sampled over an 18-year period (1995–2012) (Fig. 1A, D), from which we build time-resolved phylogenies (Fig. 1E–H and Supplementary Figs. 2–5)[11]. In Bangkok, the home location of the cases was also geocoded ($N = 467$) (Fig. 1B). To inform our model, we use data from a major mobile phone operator in Thailand that empirically captures how adults move ($N = 11.4$ million subscribers; 26% market share), modelled estimates of the probability of *Aedes aegypti* occurrence (Supplementary Fig. 6) and the long-term spatiotemporal distribution of serotypes (Fig. 1C)[3,12]. We explore viral movement over two different spatial scales: within

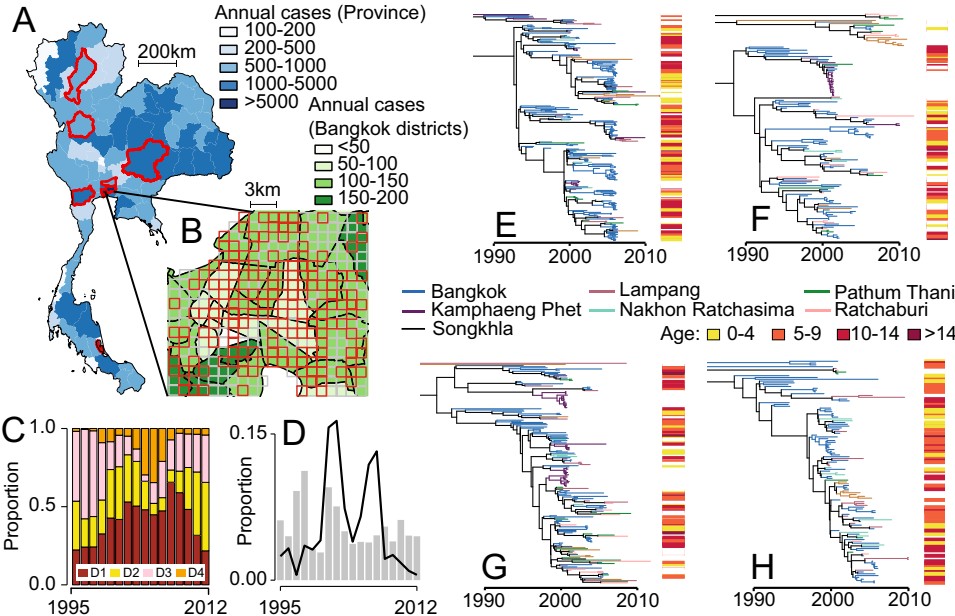

**Fig. 1 Dengue sequence and case information.** (**A**) Mean annual reported cases per province. The provinces in red represent sentinel sites where sequenced viruses are available. (**B**) Mean annual reported cases per district in central Bangkok. The grid cells in red are locations with available sequences. (**C**) Serotype distribution by year. (**D**) The distribution of all reported cases (grey) and available sequences (black line) from 1995 to 2012 by the year they were reported. Time-resolved phylogenetic trees for (**E**) DENV1, (**F**) DENV2, (**G**) DENV3 and (**H**) DENV4 with location and age information.

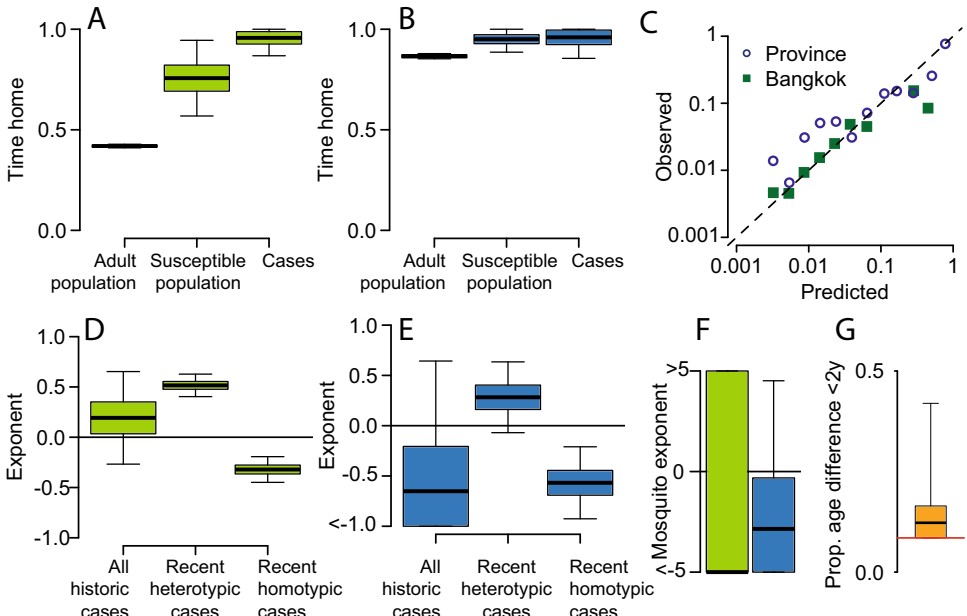

**Fig. 2 Model estimates. A** Estimated proportion of hours spent within an individual's home 1-km² grid cell in Bangkok for adults, susceptible individuals and cases. **B** Estimated time within an individual's home province. **C** Comparison of predicted and actual observed location of viruses from 20% held-out locations (where sequences from held out locations are not included in the model fitting process). **D, E** Impact of the overall number of cases throughout the study period and recent (defined as in the previous 1–2 years) homotypic and heterotypic cases on the flow of virus to any location at any time. **F** Impact of the estimated probability of *Aedes aegypti* occurrence on the probability of infection within any location (green—Bangkok, blue—provinces). **G** Proportion of infector–infectee pairs being within 2 y in age. The red line represents homogeneous mixing. The boxplots in panels (**A**, **B**) and (**D**–**G**) represent the mean estimate, with the bounds of the box representing 25th and 75th percentiles and the minima and maxima representing 95% bootstrap confidence intervals.

central Bangkok ($N = 337$ 1-km² grid cells throughout the centre of the city) and nationwide ($N = 76$ provinces with a mean area of 6700 km² each).

## Results and discussion

**Mobility of susceptible and infected populations.** Using our framework, we estimate that in Bangkok, susceptible individuals spend 76% (95% CI, 57–95%) of their time within their home cell as compared to 42% for adults (95% CI, 41–43%) (Fig. 2A and Supplementary Figs. 7–9). As dengue susceptibility is concentrated in children (Supplementary Fig. 10), our findings of reduced mobility in susceptible individuals suggest that children are less likely to travel far from their home than adults. To explore the consistency of this finding with observed differences in mobility by age, we use data from a separate study from Thailand that asked individuals ($N = 2011$) of all ages about their daily travel. Consistent with our findings of reduced mobility in susceptible individuals, we find that there is a strong relationship between age and reporting having stayed within 1 km of their home in the prior week (Supplementary Fig. 10). Incorporating the probability of being susceptible by age suggests that susceptible individuals are 1.5 (95% CI: 1.2–1.9) times as likely to report staying within 1 km of their home in the last 7 days, consistent with the 1.8 (95% CI: 1.3–2.2) times difference estimated by our model (Supplementary Fig. 10).

Using our model, we find that infected individuals in Bangkok are even less mobile than susceptible individuals with 96% (95% CI: 87–100%) of infected individuals' time being within their home cell (Fig. 2A). Importantly, these estimates of differential mobility hold for the intervening unseen transmission events, as well as the observed cases in the phylogeny. This shows that on average, infected individuals are more likely to stay in and around their home. This suggests that some subclinical DENV infections may

still result in severe enough symptoms to change the daily routine and limit mobility. Further observational studies are needed to understand how movement changes across the spectrum of disease severity[13]. We also observed similar differences in mobility patterns at the national scale, with cases spending 96% of the time within their home province (95% CI: 86–100%), compared to 95% for the susceptible population (95% CI: 89–100%) and 87% for adults (95% CI: 86–88%) (Fig. 2B).

**Role of local immunity, vector and age.** Local serotype-specific immunity also appears important, with transmission being more likely to occur in places that have seen increases of other (heterotypic) serotypes circulating in the previous two years and less likely to occur in places with increased cases of the same (homotypic) serotype within the same timeframe (Fig. 2D, E and Supplementary Figs. 8 and 9). However, the overall incidence of reported cases in the home cell or province of the susceptible population is not associated with differences in transmission risk, highlighting the complex relationship between observed case incidence and underlying infection risk. More direct measures of local immunity through population-representative seroprevalence studies may provide a more nuanced picture of the role of immunity in patterns of spread[14,15].

We find that the probability of *Aedes aegypti* presence is not linked to transmission risk (Fig. 2F and Supplementary Figs. 8 and 9), although, for Bangkok, this may be driven by limited heterogeneity in estimated presence across the city (Supplementary Fig. 6). This does not rule out a role for the vector in characterising heterogeneity in risk. In particular, the relationship between modelled probabilities of occurrence that we have used and vector density remains unclear. After accounting for age-specific patterns of immunity, we find no evidence of age-dependent transmission, with 0.12 (95% CI: 0.09–0.42) of sequential infections in a

transmission chain occurring between individuals of <2 y in age difference, which is not statistically different to models with no age structure in transmission, suggesting that the intermediary vector removes the effect of assortative mixing (Fig. 2G and Supplementary Fig. 11).

**Characterising model fit**. In order to assess the performance of our model, we repeatedly refit the model on data where we remove all sequences from a subset of locations (held-out locations). We find our model is able to accurately estimate the probability of observing viruses in the held-out locations, both within Bangkok and at the nationwide scale (correlation between observed and estimated locations of held-out sequences of 0.94 in Bangkok and 0.94 nationwide) (Fig. 2C). This demonstrates our framework can characterise viral movement in unobserved locations and highlights how non-representative sequencing approaches can still provide accurate descriptions of overall virus mobility. However, sampling biases do need to be explicitly incorporated as assuming unbiased observation results in very different parameter estimates, including falsely high estimates of between-location population movement (Supplementary Fig. 12). Using a simulation framework, we show we are able to accurately recover known parameter values, even under biased observation (Supplementary Fig. 13).

**Simulating transmission across spatial scales**. We use our fitted model to characterise the movement of the virus at each transmission generation. We use a simulation approach that

introduces viruses into randomly selected provinces and use the fitted mobility matrices to see where transmission occurs over 20 transmission generations. Averaging over repeated simulations, we find that the virus is 4.3 (95% CI: 2.4–7.0) times as likely to have travelled to Bangkok after a single-transmission generation as compared to a randomly selected province. After 20 transmission generations (equivalent to ~1 year of sequential transmissions), we find the virus is 11.4 (95% CI: 6.3–19.4) times as likely to have infected at least one individual in the capital as compared to at least one individual living in a randomly selected province (Fig. 3A and Supplementary Fig. 14). The flow to larger cities is not restricted to Bangkok. with the likeliest destinations after 20 generations also being where the largest population centres are located (Fig. 3A). Substantial heterogeneity is also observed at the local scale, with the virus tending to go to the hyper-urban city centre in Bangkok (Fig. 3B). Overall, within Bangkok, we find that 34% of infections occur outside the 1-km$^2$ home grid cell of an infected individual (95% CI: 26–43%). This is despite infected individuals spending only an average of 4% of their time outside their home cells, highlighting the importance of considering mobility in both infected and susceptible populations when considering viral spread (Fig. 3C). After 20 generations, only 2.6% of viruses are still within the same Bangkok cell and 34% are within the same province (Fig. 3D).

While most transmissions occur within the home cell of an infected individual, we find that when transmissions do occur further away, local heterogeneity in patterns of serotype-specific population immunity means that the pathway taken by viruses

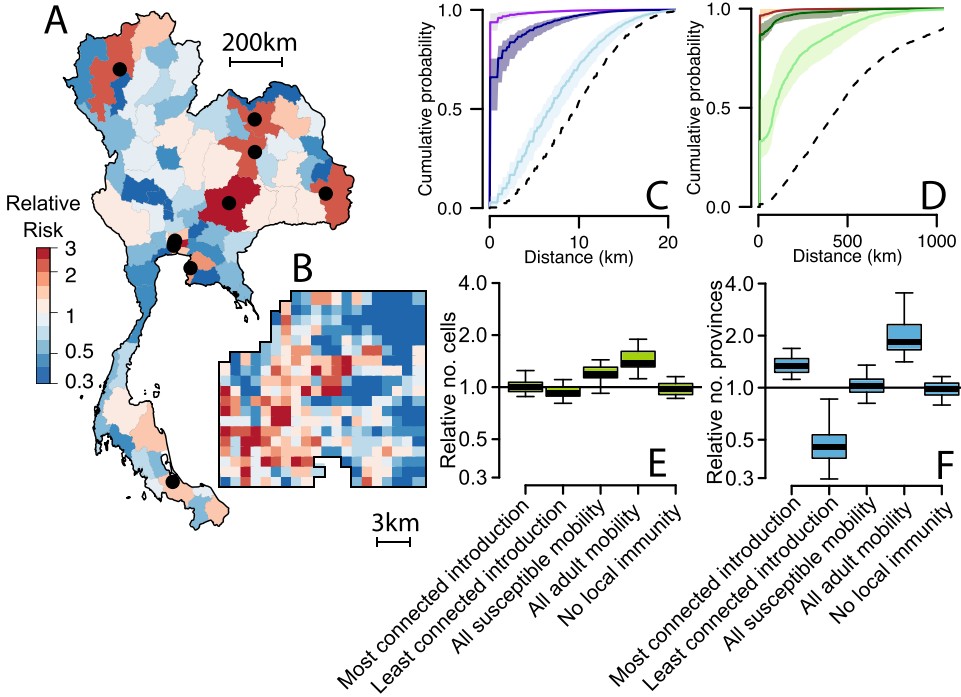

**Fig. 3 Mobility across spatial scales. A** Relative risk of movement of virus to each province compared to moving to a randomly selected province after 20 generations. The black dots represent the ten largest cities in Thailand. **B** Relative risk of movement of virus to each grid cell within central Bangkok after 20 generations compared to moving to a randomly selected cell. The cumulative distance distribution for national (**C**) and central Bangkok (**D**) of how far cases are from home (purple/brown), the distances of single transmissions (dark blue/dark green) and distances after 20 transmissions (light blue/light green). The dashed line represents the cumulative distribution function of completely spatially random movement. **E** The number of Bangkok locations with at least one case for different scenarios relative to that in the base model after 20 generations. The different scenarios are: initial introductions are in the most connected location, initial introductions are in the least connected location; no reduced mobility in cases compared to the susceptible population, case/susceptible population mobility is equal to adult mobility and no impact of local immunity on where transmission occurs. **F** For the same scenarios, the number of provinces with at least one case relative to that in the base model after 20 generations. The boxplots in panels (**E**) and (**F**) represent the mean estimate, with the bounds of the box representing 25th and 75th percentiles and the minima and maxima representing 95% bootstrap confidence intervals.

depends on the serotype (Supplementary Fig. 15). Within Bangkok, on average, 85% (95% CI: 81–89%) of the likeliest location after a single-transmission event was the same across serotypes dropping to only 44% (95% CI: 38–53%) overlap after 20 generations. These effects are not observed at a larger scale, where the likeliest destination province remained largely the same across serotypes.

Using this same simulation approach, we explore how far a virus will have spread a year following the introduction in a randomly selected location. We find that the virus will have infected individuals from, on average, 27% of all provinces (95% CI 15–38) and in 32% of cells within central Bangkok (95% CI: 25–43) (Supplementary Fig. 16). We find that local immunity and the reduced mobility of cases compared to the susceptible population has minimal effect on the number of locations affected; however, if the mobility of susceptible individuals matched that of the adult population, there would be 1.9 times as many infected provinces (95% CI: 1.4–3.3), with a similar effect at the local Bangkok scale (RR: 1.5, 95% CI: 1.1–1.9) (Fig. 3E, F). For arboviruses, such as Zika and chikungunya viruses, where limited immunity means most infections are in adults, we could therefore assume a more rapid dispersal of the virus compared to DENV[16]. We observe consistent patterns across different effective reproductive numbers and for overdispersed transmission (Supplementary Figs. 16 and 17).

**Summary**. By explicitly characterising the mechanisms of individual transmission generations and integrating the mobility of populations, our framework brings inference to a tractable scale and allows unbiased inferences to be made despite minimal and heavily biased sequence availability. Individual transmission generations are also those most relevant for targeted interventions and can help predict future flows. While we have used this framework for DENV, it is applicable to other communicable pathogens where there exists a time-resolved phylogeny, the generation time distribution is known and is relatively short (days or weeks) and there exists spatial information or other discrete traits.

## Methods
### Data sources
*Sequence data and associated metadata.* We use all available full genome sequence data from all four serotypes from Thailand covering the period 1994 and 2012 where province-level spatial information is available. All sequences are available from GenBank. The accession numbers are set out in Supplementary Data 1. For sequences from Bangkok, we also have household coordinates for a subset of 432 sequences. In Bangkok, the sequences come from Queen Sirikit National Institute of Child Health, a large children's tertiary care hospital based in the centre of the city. Outside Bangkok, most sequences come from the national surveillance system of dengue run by the Ministry of Public Health. They perform confirmatory testing and viral isolation and sequencing of samples from sentinel hospitals based around the country. The hospitals are based in the following five provinces: Lampang, Ratchaburi, Songkhla, Pathum Thani, Nakhon Ratchasima. In addition, there are sequences available on GenBank from Kamphaeng Phet province.

*Case data.* All cases of dengue are notifiable and are reported to the Thai Ministry of Public Health. We extract the number of cases per year for each year and each of the 76 provinces in the country between 1994 and 2012. To estimate serotype-specific case data in each year, we used the serotype distributions of geocoded cases from QSNICH for Bangkok (N = 11,583 cases). For the rest of the country, we used serotype-specific case data from the Ministry of Public Health. Outside the five sentinel surveillance sites, these data mainly come from ad hoc samples sent to the ministry for testing. Altogether, this represents a serotype-specific database of 27,586 cases covering 67 of the 76 provinces. For each province and year, we calculated the proportion of cases that were caused by each serotype. Where there were no samples from that province-year, we used data from the closest province in that same time period where data were available.

*Population data.* In Bangkok, we initially placed a 1 × 1-km grid cell over the central part of the city (337 grid cells) and estimated the population size using population-size estimates from LandScan for 2010[17]. We also identified the grid cell for each of the Bangkok sequences. At the province level, we used population-size estimates from the 2010 national census. We note that the Thai population has been relatively stable over the study period (rising from 60 million to 64 million between 1995 and 2012).

*Call detail records (CDR).* To estimate adult mobility across provinces and within Bangkok, we used the call detail records (CDR) of over 11 million mobile phone subscribers between August 1, 2017 and October 19, 2017 from the third-largest mobile phone operator in Thailand (N = 11.4 million subscribers, 26% market share). These data are described in more detail elsewhere[18]. Briefly, each subscriber was assigned a daily home location based on their most frequently used cellular tower. Travel between locations was estimated by tabulating the subscriber's home location on 1 day relative to the day before. The location-to-location transition probability matrix was estimated by using the average travel from location $i$ to location $j$ (weighted by the population at location $i$), and normalising travel such that the sum of travel from location $i$ is equal to 1. We note that the mobile phone data were collected after our study period (2017 vs 1995–2012). Human mobility may have changed and could help explain some of the differences in mobility between the fitted models and that implied from the mobile phone data.

*Aedes aegypti abundance estimates.* We used previously published estimates of the probability of *Aedes aegypti* presence for 5 × 5-km grid cells around the globe[19]. These estimates were generated by incorporating information on temperature, rainfall, vegetation indices from satellite imagery and fitting models to a large dataset of *Aedes* occurrence records. The fitted models were then used to predict elsewhere. For Bangkok, we extracted the *Aedes aegypti* estimate using the centroid of each grid cell. For the province level, for each province, we used the simple average across all the raster cells from the *Aedes* map that were contained within that province.

**Generation time distribution for dengue.** To estimate the generation time distribution for dengue, we combined data on the incubation period, extrinsic incubation period and the lifespan of the *Aedes aegypti* mosquito as has previously been used for chikungunya[20].

*Human incubation period (HI).* We used a truncated log-normal distribution with a mean of 5.6 days and a standard deviation of 1.41 days and a maximum time of two weeks[21]. The stated mean and standard deviation are the values prior to truncation.

*Human-to-mosquito transmission (HM).* Based on the estimated durations of viremia, we used a truncated exponential distribution with a mean of 4.5 days and a maximum period of 7 days[22].

*Mosquito infectiousness (MI).* The period of mosquito infectiousness depends on the mosquito lifespan and the extrinsic incubation period. The average daily probability of survival for *Aedes aegypti*, has been estimated at 0.87 for up to 30 days[23], equivalent to a mean lifespan of 7.2 days. The extrinsic incubation period has been estimated at 6.1 days[24]. To calculate the period of mosquito infectiousness, we initially draw the mosquito lifespan (MLS) using a truncated exponential distribution with a parameter of 7.2 days and a maximum value of 30 days. Next, we draw the age at which the mosquito gets infected (MAI) from a uniform distribution between 0 and the lifespan of the mosquito. Next, we draw the extrinsic incubation period (EIP) as a random exponential distribution with mean of 6.1 days. The total period of mosquito infectiousness (MI) is then equal to MLS–MAI–EIP. Values of MI <0 were considered unsuccessful onward infections.

*Generation time distribution.* We derived the empirical distribution of the generation time by simulating 10,000 values for HI, HM and MI and summing them. Individuals who are viremic for longer are more likely to infect mosquitoes. Similarly, mosquitoes that are infectious for longer are also more likely to infect more individuals. We, therefore, weighted the probability of each generation time by the length of HM multiplied by the length of MI. We obtained a mean generation time of 18.2 days and a standard deviation of 6.1 days, which we approximated using a gamma distribution with the same mean and standard deviation (Supplementary Fig. 1).

**Time-resolved phylogenetic trees.** Taking each serotype in turn, we aligned the full genome sequences using the Muscle algorithm in MEGA[25]. We built Bayesian time-resolved phylogenetic trees using BEAST 2.5.0[11]. We used a strict clock, a General Time Reversible nucleotide substitution model, as determined by jModelTest2[26], and a Bayesian skyline prior. Similar coalescence times were found using a relaxed clock.

**Probabilistic model**
*Overall inferential strategy.* We use a likelihood-based approach to model the probability of the observed location of pairs of sequences in a time-resolved phylogeny. We initially use the time-resolved phylogenies and information on the generation time distribution to estimate the number of generations that separates

each member of a pair of sequences in the phylogeny from their Most Recent Common Ancestor (MRCA). This then allows us to consider single-transmission generations rather than overall viral flow. We develop models of viral movement between each location in our study area (whether it was sampled or not) for each transmission step. These viral movement models incorporate estimates of human mobility with the potential for differences in the movement for cases compared to the susceptible population, as well as incorporating effects of time-varying serotype-specific immunity and vector distributions. We allow for infection to occur at the infector's home location (which requires the infectee travelling to the infector's home location), the infectee's home location (which requires the infector to travel to the infectee's home location) or in a tertiary location (where both parties would have to travel there).

We use the viral movement matrix for a single transmission to calculate the viral movement after $G$ generations via matrix multiplication. This approach integrates over all possible pathways that link two locations. To specifically incorporate observation processes, we consider the probability of sequencing an infection at each space–time unit.

To inform this model, we use an integrative approach that brings in detailed data from mobile phone operators that capture how people move and interact with each other, maps that estimate how populations are distributed, maps on vector suitability and the long-term spatiotemporal distribution of serotypes.

*Notation.* For a pair of cases, $C_A$ and $C_B$, in a phylogenetic tree: $C_A$ has home location $L_A$, was sick at time $T_A$ and has sequence $Seq_A$; case $C_B$ has home location $L_B$, was sick at time $T_B$ and has sequence $Seq_B$. The time of the MRCA between $C_A$ and $C_B$ is $T_m$ and the location of the MRCA is $L_m$. We denote $G_A$ and $G_B$ the number of transmission generations that separate $C_A$ and $C_B$ from their MRCA. $Obs_{LiTA}$ is 1 if a sequence was observed at location $Li$ at time $T_A$ and 0 otherwise. $Obs_{LiTB}$ is defined in a similar manner for time $T_B$.

*The single-transmission generation matrix.* Initially let us consider a single-transmission generation. The probability that two individuals, $i$ and $j$, are in the same location and are in contact (via a mosquito) given $i$ lives in location $a$ and $j$ lives in location $b$ can be written down as

$$P(\text{persons } i \text{ and } j \text{ are in contact} | L_i = a, L_j = b) = \sum_k P(V_i = k | L_i = a) \cdot P(V_j = k | L_j = b) \cdot \beta_k$$
(1)

where $P(V_i = k | L_i = a)$ is the probability of individual $i$, whose home location is in $a$, visiting location $k$ and $P(V_j = k | L_j = b)$ is the probability that individual $j$ also visits location $k$ and $\beta_k$ is the location-specific probability of transmission.

At time $\tau$, one infector $i$ that lives in location $a$ is expected to transmit to the following number of persons living in location $b$:

$$E(\text{number of persons from } b \text{ infected at time } \tau | L_i = a)$$
$$= \sum_k P(V_i = k | L_i = a) \cdot P(V_j = k | V_j = k, L_j = b) \cdot \beta_k \cdot S_{b, \tau, \text{ser}}$$
(2)

where $S_{b, \tau, \text{ser}}$ is the number of susceptible people to serotype *ser* living in location $b$ at time $\tau$. The total expected number of persons infected by the infector is $i$:

$$E(\text{number of persons from all locations infected at time } \tau | L_i = a)$$
$$= \sum_m^N \sum_k^N P(V_i = k | L_i = a) \cdot P(V_j = k | L_j = m) \cdot \beta_k \cdot S_{m, \tau, \text{ser}}$$
(3)

Conditional on transmission occurring, the probability that the infectee has a home location in cell $b$ is the ratio of these terms:

$$\pi_{a,b,\tau,\text{ser}} = P(L_j = b | L_i = a) = \frac{\sum_k^N P(V_i = k | L_i = a) \cdot P(V_j = k | L_j = b) \cdot \beta_k \cdot S_{b,\tau,\text{ser}}}{\sum_m^N \sum_k^N P(V_i = k | L_i = a) \cdot P(V_j = k | L_j = m) \cdot \beta_k \cdot S_{m,\tau,\text{ser}}}$$
(4)

We can create a $N \times N$ transmission matrix, $\prod_{\tau,\text{ser},\text{gen}=1}$, where $N$ is the total number of locations, that sets out the transmission probabilities between all pairs of locations at a point in time for a single transmission. The element $[a,b]$ of the matrix is $\pi_{a,b,\tau,\text{ser}}$.

*Characterising human mobility.* We use mobile phone data to characterise human mobility. Initially, we extract a matrix from the CDR data that set out the probability that an individual that lives in location $a$ visits location $k$.

CDRs come from adults, whereas dengue is concentrated in children, who are potentially more likely to spend more of their time at home. In addition, sick individuals may travel differently than healthy individuals and spend more time at home. In this way, the mobility of infectors may differ from susceptible information.

To allow for different periods at home for susceptible individuals compared to those in the CDR data and to assess whether there exists differential mobility by illness status, we incorporate separate parameters for the probability of being at home for infectors and the susceptible population ($\theta_{\text{infector.home}}$, $\theta_{\text{population.home}}$) to reflect the additional time at home compared to that extracted from the CDR data.

For the home cells of infected individuals:

$$P(V_i = a | H_i = a) = \min(0.999, \text{CDR}[a,a] \cdot \theta_{\text{infector.home}})$$
(5)

For the home cells of the rest of the population:

$$P(V_i = a | H_i = a) = \min(0.999, \text{CDR}[a,a] \cdot \theta_{\text{population.home}})$$
(6)

In each case, for non-home cells, we rescale the probabilities so that the sum of all movements remains equal to 1.

$$P(V_i = k | H_i = a, k \neq a) = \text{CDR}[a,k] / (1 - P(V_i = a | H_i = a))$$
(7)

where CDR[a,k] reflects the movement probabilities from the CDR data.

As the sum of movements to the destinations in the matrix is equal to 1, we are assuming that the spatial unit of analysis contains all possible mobility (of the virus and people). It has previously been shown that the dengue epidemic in Thailand is self-sustaining with few external introductions[3]. Applications of this approach to small spatial units should consider that some mobility may be missed.

Dengue transmission is more likely to occur during daylight hours, due to the feeding behaviour of *Aedes* mosquitoes. Therefore, it would be optimal to use CDR data from daylight hours only. However, as is often the case, our CDR data represent an aggregate from all hours of the day. Nevertheless, as the majority of cell phone calls are made during daylight hours (it has been estimated that 67% of calls are between 8 am and 7 pm)[27,28]—it is reasonable to assume that this estimate is largely representative of daytime mobility.

*Factors affecting transmission.* As there may be factors that allow for different probabilities of transmission across locations, we allowed for differential probability of infection by location based on the mosquito presence in that location

$$\beta_k = D_1 \cdot \text{mosq}_k^{\gamma_1}$$
(8)

where parameter $\gamma_1$ is to be estimated, mosq represents the estimated suitability for *Aedes aegypti* mosquitoes in location $k$. $D_1$ is a proportionality constant (which gets cancelled out).

*Factors affecting susceptibility.* The number of susceptible individuals living in a location will depend on the level of historic infection in that location, in a potentially time-varying serotype-specific manner.

$$S_{k,\tau,\text{ser}} = D_2 \cdot \text{het}_{k,\tau,\text{ser}}^{\beta_1} \cdot \text{homo}_{k,\tau,\text{ser}}^{\beta_2} \cdot \text{incidence}_k^{\beta_3}$$
(9)

where $\text{het}_{k,\tau,\text{ser}}$ is the incidence of cases caused by different serotypes in the two prior years in location $k$, $\text{homo}_{k,\tau,\text{ser}}$ is the incidence of cases caused by the same serotype in the two prior years in location $k$, $\text{incidence}_k$ is the incidence of all cases over the study period in location $k$. We choose a window of 2 years to define recent immunity as serotype-specific incidence has previously been shown to be spatially correlated over this time range, presumably due to serotype-specific local herd immunity[9]. For the Bangkok analyses, we use the serotype-specific geolocated case data. For the nationwide analysis, we use the national reporting system from the Ministry of Public Health (MOPH). The national MOPH system is not serotype-specific; however, a small number of cases from all around the country are serotyped each year by the national reference centre in Bangkok. To obtain serotype-specific incidence estimates for each year, we multiplied the proportion of cases that came from each serotype within each province-year by the overall number of cases for each province-year. Where there were no serotyped cases for a province-year, we used the closest province where there were serotyped cases.

*Probability of virus being within each location after G transmission generations.* To calculate the probability of the home location being within location $k$ after $G$ transmission generations, we can use matrix multiplication that integrates over all possible pathways connecting two locations

$$\prod_{\tau=T_G, \text{ser}, \text{gen}=G} = \prod_{l=2}^{G} \prod_{\tau=t_{l-1}, \text{ser}, \text{gen}=l-1} \prod_{\tau=t_l, \text{ser}, \text{gen}=1}$$
(10)

where $t_l$ is the time of generation $G_l$.

*Probability of observing a pair of cases in two specific locations.* Conditional on sequences being observed in location $L_A$ at time $T_A$ and $L_B$ at time $T_B$, the probability that $C_A$ has home location $L_A$ and $C_B$ has home location $L_B$ can be written down as

$$P(L_A, L_B | \text{Obs}_{L_A, T_A}, \text{Obs}_{L_B, T_B}, \text{Seq}_A, \text{Seq}_B, T_A, T_B)$$
$$= \frac{P(\text{Obs}_{L_A, T_A}, \text{Obs}_{L_B, T_B} | L_A, L_B, \text{Seq}_A, \text{Seq}_B, T_A, T_B) P(L_A, L_B | \text{Seq}_A, \text{Seq}_B, T_A, T_B)}{\iint_{L_i, L_j} P(\text{Obs}_{L_i}, L_A, \text{Obs}_{L_j, T_B} | L_A, L_B, \text{Seq}_A, \text{Seq}_B, T_A, T_B) P(L_i, L_j | \text{Seq}_A, \text{Seq}_B, T_A, T_B) dL_i dL_j}$$
(11)

We can consider that the location of the two cases is dependent on the location of their MRCA and the number of transmission generations that separate them

from the MRCA.

$$P(L_A, L_B | \text{Seq}_A, \text{Seq}_B, T_A, T_B)$$
$$= \iiint_{L_m, G_A, G_B} P(L_A, L_B | L_m, G_A, G_B) \cdot P(L_m) \cdot P(G_A, G_B | \text{Seq}_A, \text{Seq}_B, T_A, T_B) dL_m dG_A dG_B \quad (12)$$

We consider that the observation processes across locations are independent of each other. In addition, each transmission event is considered independent of other transmission events. The probability of observing a case at location $L_i$ at time $T_A$ does not depend on the location of the MRCA or the number of generations separating the case from the MRCA. We can also substitute in Eq. (12) into Eq. (11). Finally, we consider discretized space—either 337 1 ×1-km grid cells throughout central Bangkok or the 76 provinces of Thailand.

Equation (11) therefore becomes

$$P(L_A, L_B | \text{Obs}_{L_A, T_A}, \text{Obs}_{L_B, T_B}, \text{Seq}_A, \text{Seq}_B, T_A, T_B)$$
$$= \frac{\sum_{L_m} \sum_{G_A} \sum_{G_B} P(\text{Obs}_{L_A}, T_A) P(\text{Obs}_{L_B, T_B}) P(L_A | L_m, G_A) P(L_B | L_m, G_B) P(L_m) P(G_A, G_B | \text{Seq}_A, \text{Seq}_B, T_A, T_B)}{\sum_{L_i} \sum_{L_j} \sum_{L_m} \sum_{G_A} \sum_{G_B} P(\text{Obs}_{L_i, T_A}) P(\text{Obs}_{L_j, T_B}) P(L_i | L_m, G_A) P(L_j | L_m, G_B) P(L_m) P(G_A, G_B | \text{Seq}_A, \text{Seq}_B, T_A, T_B)}$$
$$(13)$$

*Probability of G generations between the MRCA and a case.* We can extract the joint probability that case $C_A$ is separated from the MRCA by $G_A$ transmission generations and case $C_B$ is separated from the same MRCA by $G_B$ transmission generations using the generation time distribution, for dengue and the time-resolved phylogenetic tree.

If we assume that the generation time distribution is gamma distributed with parameters $a_G$ and $\beta_G$ and that all transmission events are independent of each other, the sum of g gamma distribution is also gamma distributed with parameters $g\alpha_G$ and $\beta_G$. In addition, from a genealogy, $R_i$, we can extract the evolutionary time, $E_A$, separating $C_A$ from the MRCA and $E_B$, separating $C_B$ from the MRCA. We can therefore estimate the probability of g transmission events over many trees as follows:

$$P(G_A, G_B | \text{Seq}_A, \text{Seq}_B, T_A, T_B) = \frac{\sum_l f(E_{A,R_l}; G_A \cdot \alpha_G, \beta_G) \cdot f(E_{B,R_l}; G_B \cdot \alpha_G, \beta_G)}{\sum_i \sum_l \sum_m f(E_{A,R_l}; l \cdot \alpha_G, \beta_G) \cdot f(E_{B,R_l}; m \cdot \alpha_G, \beta_G)} \quad (14)$$

This approach allows us to incorporate uncertainty in the phylogeny, including uncertainty in the evolutionary parameters and tree structure.

As any spatial signal will be heavily diluted after many transmission generations, to optimise computational performance, we restrict our analyses to pairs where the mean estimated number of transmission generation is <25, we perform a sensitivity analysis where this is extended to 40 generations with very similar results (Supplementary Fig. 3).

*Observation probability (P(Obs$_{Li,TA}$)).* We cannot know the true number of infections occurring within each space–time unit. Given the long-term endemicity of dengue in the region, we assume that the number of infections will be approximately proportional to the size of the population within each location. Therefore, the probability of observation (the probability of sequencing the virus causing an infection event) at location $k$ at time point $t$ is approximately proportional to the number of sequenced viruses from that year and location for that serotype divided by the size of the population in that location.

We conducted a sub-analysis where we assumed unbiased observation. In this analysis, we assumed that the probability of observation was 1 across all space–time locations, we obtained very different results (Supplementary Fig. 6).

We further assessed the performance of this approach using a simulation model that imposed a heavily biased observation process. Our inference framework was able to correctly identify all parameters (Supplementary Fig. 7, see below for simulation model details).

*The location of the MRCA (P(L$_m$)).* The probability of the MRCA for each pair of cases will depend on the long-term history of dengue in the communities, which cannot be estimated using the presented approach. Instead, we assume that $P(L_m)$ is proportional to the size of the population in that location. To assess the sensitivity of this assumption, we conducted a separate analysis where $P(L_m)$ was assumed to be the same across all locations, with identical results (Supplementary Fig. 4). This suggests that we do not need to probabilistically assess where the start point is for the MRCA that links two cases in a phylogeny.

*Likelihood.* We can calculate the likelihood using all pairs of available sequenced viruses as follows:

$$L \propto \prod_{\text{ser}=1}^{4} \prod_{i=1}^{n_{\text{ser}}} \prod_{j \ne i} P(L_i, L_j | \text{Obs}_{Li, Ti}, \text{Obs}_{Lj, Tj}, \text{Seq}_i, \text{Seq}_j, T_i, T_j) \quad (15)$$

where $n_{\text{ser}}$ are the number of sequences available from serotype ser.

*Identifying the maximum likelihood estimate.* We use a maximum likelihood approach to estimate the parameters linked to the mobility, transmission and susceptibility ($\theta_{\text{home.sick}}$, $\theta_{\text{home-population}}$, $\beta_1$, $\beta_2$, $\beta_3$, $\gamma_1$). We identify the maximum likelihood estimate using an unconstrained nonlinear quasi-Newton optimisation approach[29].

In order to incorporate uncertainty, we use a bootstrapping approach where we randomly sample all the available sequences with replacement over 100 iterations

and recalculate the maximum likelihood estimate for each parameter each time. The 95% confidence intervals are then calculated using the mean and the standard deviation of the resulting distribution, assuming that they follow a normal distribution.

*Using fitted values to estimate patterns of viral flow at each transmission generation.* Once we have fitted values for the parameters, we can calculate the $\Pi_{r, \text{ser}, \text{gen}}$ matrix for each month between 1994 and 2012, each serotype and each transmission generation. From these matrices, we can extract the probability that the virus is in each location given a specified serotype, location and time of introduction and number of generations. From these matrices, we calculate the cumulative distribution function of the distance between where the virus started and where it is after different numbers of generations, averaging over time and serotype. We compare this to the cumulative distribution function of how far cases are from their home at any time. This highlights that viral mobility requires both movement of cases and the susceptible population.

We also calculate the mean proportion of times that the most likely (non-home) destination is the same across serotypes. This allows us to assess whether viruses across the serotypes take the same routes or whether serotype-specific immunity changes the most likely pathways.

*Model fit.* In order to assess the model fit, we perform held-out validation. In Bangkok, we remove all sequences from 10% randomly chosen locations, we then refit the model and then estimate the probability of observing sequences in the locations not included in the model fitting process. For the nationwide analysis, as we have fewer locations with sequences available, we undertake the same process but hold out a single province in turn.

For incremental windows of probability between 0 and 1, we identify all location-years where a virus was predicted to have been observed within the held-out locations. We then calculate the mean proportion of times a sequence was observed within those identified location-years.

*Estimates of spread using a transmission simulation.* Using the fitted parameter values, we conduct a forward simulation at both the Bangkok and province levels. Taking each month between 1994 and 2012 and each serotype in turn, we apply the following algorithm:

(I) Randomly introduce a single infection in one location where all locations have the same probability of being the source.
(II) We generate daughter infections from the index using a random draw from a Poisson distribution with mean $R_{\text{eff}}$ (representing the effective reproductive number).
(III) We identify the location for each daughter infection using a random draw where the probability of each location is taken from the $\Pi_{r, \text{ser}, \text{gen}}$ matrix.
(IV) Repeat (ii) and (iii) for 20 generations.
(V) Repeat (i)–(iv) 50 times.

For each iteration, we calculate the average number of locations that have had at least one infection at each generation, average over all time points and all serotypes.

To assess the impact of mobility patterns and immunity on the number of locations affected, we repeated the analysis with the following adjustments:

Scenario a: All initial introductions were in the most connected location only (as defined as the location with the lowest probability of staying within your home location).

Scenario b: All initial introductions were in the least connected location only (as defined as the location with the highest probability of staying within your home location).

Scenario c: No difference in the mobility of the cases as compared to the susceptible population. This was achieved by forcing the $\theta_{\text{infector. home}}$ parameter to be the same as the fitted $\theta_{\text{population. home}}$ parameter.

Scenario d: Susceptible population mobility is equal to that of the adult population. This was achieved by forcing the $\theta_{\text{population. home}}$ and $\theta_{\text{infector. home}}$ parameters to be zero.

Scenario e: No impact of immunity. This was achieved by forcing the $\beta_1$ and $\beta_2$ parameters to be zero.

For each scenario, we calculated the proportion of locations affected at each generation and the relative number of locations affected compared to the base model. We also conducted sensitivity analyses where the $R_{\text{eff}}$ was varied from 1.3 to 1.1 and 1.6.

**Age-mixing model.** We use an equivalent approach to characterise the age-dependent mixing of the population. Instead of considering the probability of the virus transitioning between two locations, we consider the probability of transitioning between individuals of two ages.

$$P\left(\text{Age}_A, \text{Age}_B | \text{Obs}_{\text{Age}_A, T_A}, \text{Obs}_{\text{Age}_B, T_B}, \text{Seq}_A, \text{Seq}_B, T_A, T_B\right)$$
$$= \frac{P\left(\text{Obs}_{\text{Age}_A, T_A}, \text{Obs}_{\text{Age}_B, T_B} | \text{Age}_A, \text{Age}_B, \text{Seq}_A, \text{Seq}_B, T_A, T_B\right) P\left(\text{Age}_A, \text{Age}_B | \text{Seq}_A, \text{Seq}_B, T_A, T_B\right)}{\iint_{\text{Age}_i, \text{Age}_j} P\left(\text{Obs}_{\text{Age}_i, T_A}, \text{Obs}_{\text{Age}_j, T_B} | \text{Age}_A, \text{Age}_B, \text{Seq}_A, \text{Seq}_B, T_A, T_B\right) P\left(\text{Age}_i, \text{Age}_j | \text{Seq}_A, \text{Seq}_B, T_A, T_B\right) d\text{Age}_i d\text{Age}_j}$$
$$(16)$$

We can consider that the age of the two cases is dependent on the age of the MRCA and the number of transmission generations that separate them from the MRCA.

$$P\left(\text{Age}_A, \text{Age}_B | \text{Seq}_A, \text{Seq}_B, T_A, T_B\right)$$
$$= \iint_{\text{Age}_m, G_A, G_B} P\left(\text{Age}_A, \text{Age}_B | \text{Age}_m, G_A, G_B\right) \cdot P(\text{Age}_m) \cdot P(G_A, G_B | \text{Seq}_A, \text{Seq}_B, T_A, T_B) d\text{Age}_m dG_A dG_B$$
(17)

We consider that the observation processes across ages are independent of each other. In addition, each transmission event is considered independent of other transmission events. The probability of observing a case at age $\text{Age}_i$ at time $T_A$ does not depend on the location of the MRCA or the number of generations separating the case from the MRCA. We can also substitute Eq. (12) into Eq. (11). Finally, we consider the discretized ages.

$$P\left(\text{Age}_A, \text{Age}_B | \text{Obs}_{\text{Age}_A, T_A}, \text{Obs}_{\text{Age}_B, T_B}, \text{Seq}_A, \text{Seq}_B, T_A, T_B\right)$$
$$= \frac{\sum_{\text{Age}_m} \sum_{G_A} \sum_{G_B} P\left(\text{Obs}_{\text{Age}_A, T_A}\right) P\left(\text{Obs}_{\text{Age}_B, T_B}\right) P(\text{Age}_A | \text{Age}_m, G_A) P(\text{Age}_B | \text{Age}_m, G_B) P(\text{Age}_m) P(G_A, G_B | \text{Seq}_A, \text{Seq}_B, T_A, T_B)}{\sum_{\text{Age}_i} \sum_{\text{Age}_j} \sum_{\text{Age}_m} \sum_{G_A} \sum_{G_B} P\left(\text{Obs}_{\text{Age}_i, T_A}\right) P\left(\text{Obs}_{\text{Age}_j, T_B}\right) P(\text{Age}_i | \text{Age}_m, G_A) P(\text{Age}_j | \text{Age}_m, G_B) P(\text{Age}_m) P(G_A, G_B | \text{Seq}_A, \text{Seq}_B, T_A, T_B)}$$
(18)

**The age transition matrix**. Initially let us again consider a single-transmission generation. We assume that the age-specific susceptibility of the population is stable over time and that the probability of exposure does not differ by age group or serotype. The probability that two individuals, $i$ and $j$, of ages $\text{Age}_i$ and $\text{Age}_j$ are in contact (via a mosquito) can be written down as

$$P\left(\text{persons } i \text{ and } j \text{ are in contact} | \text{Age}_i = a, \text{ Age}_j = b\right) = \beta_{a,b} \quad (19)$$

where $\beta_{a,b}$ is the age-specific probability of contact between individuals of ages $a$ and $b$.

The expected number of infected persons coming from individuals of age $b$ conditional on an infector, $i$, being of age $a$ is

$$E\left(\text{number of persons age } b \text{ infected} | \text{Age}_i = a\right) = \beta_{a,b} \cdot S_b \quad (20)$$

where $S_b$ is the number of susceptible people of age $b$.

The expected number of infected persons coming from individuals of all ages, conditional on an infector, $i$, being of age $a$ is

$$E\left(\text{number of persons of all ages infected} | \text{Age}_i = a\right) = \sum_{m}^{N} \beta_{a,m} \cdot S_m \quad (21)$$

Conditional on one transmission generation where the infector $i$ is of age $a$, the probability of the infectee with age $b$ is, therefore, the ratio of these terms. We define this probability as $\phi_{a,b}$.

$$\phi_{a,b} = P\left(\text{Age}_j = b | \text{Age}_i = a\right) = \frac{\beta_{a,b} \cdot S_b}{\sum_{m}^{N} \beta_{a,m} \cdot S_m} \quad (22)$$

We can create an $N_{\text{age}} \times N_{\text{age}}$ transmission matrix, $\Phi$, that sets out the transmission probabilities between all ages for a single transmission where element $[a,b]$ of the matrix is $\phi_{a,b}$. We use a maximum age of 70 years.

**Age contact matrix**. To parametrically characterise the age contact matrix, we use a discretised exponential decay parameter, $\theta_{\text{age}}$, that captures the probability that two people interact, as a function of the difference in their ages, such that $\beta_{a,b} = f(a - b; \theta_{\text{age}})$.

$$\beta_{a,b} = \frac{g(a - b; \theta_{\text{age}})}{\sum_{m} g(a - m; \theta_{\text{age}})} \quad (23)$$

$$\text{where } g\left(x; \theta_{\text{age}}\right) = \theta_{\text{age}} \cdot \exp(-\theta_{\text{age}} \cdot x).$$

**Age-specific susceptibility**. To characterise the susceptibility, we assume that the number of susceptible people of age $a$ is equal to $N_{\text{age}_a} \cdot \exp(-\lambda \cdot \text{age}_a)$, where $N_{\text{age}_a}$ is the number of people of age $a$ in the national census and the force of infection, $\lambda$, is assumed to be 0.04[30]. We conduct a sensitivity analysis where the force of infection is varied to 0.02 and 0.06 with unchanged results (Supplementary Fig. 5).

**Observation probability ($P(\text{Obs}_{\text{Age}_i, T_A})$)**. We assume that the probability of observing a case of age $a$ is proportional to the number of sequenced viruses of that age for that serotype divided by the estimated size of the susceptible population of that age ($S_a$).

**Likelihood for the age model**. We can calculate the likelihood using all pairs of available sequenced viruses as follows:

$$L \propto \prod_{\text{ser}=1}^{4} \prod_{i=1}^{n_{\text{ser}}} \prod_{j \neq i}^{n_{\text{ser}}} P\left(\text{Age}_i, \text{Age}_j | \text{Obs}_{\text{Age}_i, Ti}, \text{Obs}_{\text{Age}_j, Tj}, \text{Seq}_i, \text{Seq}_j, T_i, T_j\right) \quad (24)$$

where $n_{\text{ser}}$ are the number of sequences available from serotype ser.

We use a maximum likelihood approach to estimate the parameter $\theta_{\text{age}}$. In order to incorporate uncertainty, we use a bootstrapping approach where we randomly sample all the available sequences with replacement over 100 iterations and recalculate the maximum likelihood estimate for each parameter each time. The 95% confidence intervals are then calculated using the mean and the standard deviation of the resulting distribution, assuming that they follow a normal distribution.

Once we have fitted the $\theta_{\text{age}}$ parameter, we calculate from the matrix of $\phi_{a,b}$, the proportion of transmissions that are between individuals that have <2 y in age between them. We use an equal weight for the age of the infector across all ages and only consider individuals between the ages of 1 and 15 as they represent the majority of the susceptible population. We compare this to the scenario where all individuals have the same probability of contact, irrespective of age (i.e., $\beta_{a,b} = 1/70$, for all $a$ and all $b$), which is the minimum possible value.

*Simulation study*. In order to ensure that our model is able to correctly identify parameters, we built a simulation framework with known parameters using 50 randomly selected grid cells from the Bangkok dataset where the population size, the historic incidence and the probability of between cell movement were taken from the observed data. As the observed heterogeneity in mosquito presence was limited in Bangkok (Supplementary Fig. 2), we simulated mosquito presence in each location using a Uniform distribution between 0 and 1. For the recent heterotypic and homotypic cases, we used a randomly selected time point from the observed distributions of cases and assume all cases came from serotype DENV1.

We fixed the parameter values as follows:

- Additional time being at home for a susceptible population compared to adults ($\theta_{\text{susceptible. home}}$) (logit scale): $-0.5$
- Additional time being at home for cases compared to adults ($\theta_{\text{infector. home}}$) (logit scale): $-0.05$
- Mosquito exponent: 1.0
- All incidence exponents: 0.5
- Recent heterotypic incidence exponent: 0.3
- Recent homotypic incidence exponent: $-0.3$

We then calculated the $\Pi_{r,\text{ser,gen}}$ transmission matrix using these known parameters and simulated transmission events using the following algorithm:

1. Randomly select a starting location ($H_0$) by randomly choosing a location, weighted by the population in that location. This will represent the MRCA case ($C_0$) between the two observed viruses.
2. Draw the number of generations ($g$) between the MRCA and one of the observed isolates, where the number of generations is between 15 and 19 generations and the probability of 15 generations is 0.1, 16 generations is 0.2, 17 generations is 0.4, 18 generations is 0.2, 19 generations is 0.1.
3. For case $C_0$ identify where they will transmit to ($H_1$) using a random draw with the probabilities of each destination location coming from the $H_0$ row of the $\Pi_{r,\text{ser,gen}}$ matrix.
4. Repeat step (3) $g$ times using the destination of the previous step as the start location each time
5. Repeat steps 1–4 2000 times to generate 2000 pairs of cases
6. We assumed that the probability of observing (i.e., sequencing) the virus from a case was unequal across locations. The probability of observing a case at a location ($\rho_l$) is taken from a random uniform distribution ($U(0,1)$). We randomly select 500 pairs of cases where the probability of observation of each pair is the product of the probability of observation at each of the two locations.

Using the observed pairs, we then used our framework to estimate the parameters of the model. We repeated the simulation 50 times and report the mean and 2.5 and 97.5 percentiles of the distribution for each parameter estimate.

To assess the importance of incorporating sampling bias in our estimates, we repeated the inference on our simulated data but assumed that all space–time locations had the same, equal chance of being observed.

*SMILI data and analysis*. In order to understand the consistency of our estimated differences in human mobility between susceptible individuals and adults, we used data from the Social Mixing for Influenza-Like Illness (SMILI) project. This project asked 2011 individuals about their mobility patterns. Here we used the responses to the question 'what is the farthest distance you have travelled within the last 7 days?'. We dichotomised the results into those that had not travelled >1 km and those that had travelled >1 km.

To reconstruct the probability that a susceptible individual had not travelled more than 1 km within the last 7 days, we used data on the population size from the 2010 census and assumed a constant force of infection (foi) of 0.04 per year. Using a catalytic model, we can calculate the probability that an individual of age $a$ has never been infected by dengue as $p\text{naive}_a = \exp(-4 \cdot a \cdot \text{foi})$. The probability of being monotypically immune (i.e., being infected by one of the serotypes but still susceptible to one of the other ones) is $p\text{mono}_a = \exp(-3 \cdot \text{foi} \cdot a) \cdot (1 - \exp(-\text{foi} \cdot a))$. The probability of being susceptible as the sum of $p\text{sus}_a = p\text{naive}_a + p\text{mono}_a$ (Supplementary Fig. 3B). To calculate the average probability that a

susceptible individual has not travelled further than 1 km from their home within the last 7 days, we take a weighted average across all ages:

$$pTravel_{sus} = \sum_a pTravel_a \cdot pSus_a \cdot pPop_a / \sum_a pSus_a \cdot pPop_a \quad (25)$$

where $pTravel_a$ is the proportion of individuals of age $a$ that have not travelled more than 1 km within the last 7 days and $pPop_a$ is the proportion of the population that is of age $a$.

To calculate the average probability that an adult has not travelled further than 1 km from their home within the last 7 days, we use a similar approach where we calculate a weighted average across all individuals that are over 15 years of age (the mobility data are available in 5-year increments).

*Ethical approval.* This study was approved by the ethical review boards of Queen Sirikit National Institute of Child Health, and Walter Reed Army Institute of Research and the University of Florida. Case data were obtained from the results of standard confirmatory testing for dengue and therefore did not require informed consent.

**Reporting summary.** Further information on research design is available in the Nature Research Reporting Summary linked to this article.

## Data availability
All data used in the analyses are available on Zenodo (https://doi.org/10.5281/zenodo.4543279). In addition, GenBank references for the sequences are available in Supplementary Data 1. LandScan data are available from https://landscan.ornl.gov/landscan-datasets.

## Code availability
R code used for the analyses is available on Zenodo (https://doi.org/10.5281/zenodo.4543279).

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

## Acknowledgements
H.S. is funded by the European Research Council (No. 804744). H.S. and D.A.T.C. would like to recognise funding by The National Institutes of Health (R01AI114703). A.P.W. is funded by a Career Award at the Scientific Interface by the Burroughs Wellcome Fund, by the National Library of Medicine of the National Institutes of Health under Award Number DP2LM013102 and the National Institute of Allergy and Infectious Diseases of the National Institutes of Health under Award Number R21AI151750. The content is solely the responsibility of the authors and does not necessarily represent the official views of the National Institutes of Health. Material has been reviewed by the Walter Reed Army Institute of Research. There is no objection to its presentation and/or publication. The opinions or assertions contained herein are the private views of the author, and are not to be construed as official, or as reflecting true views of the Department of the Army or the Department of Defense.

## Author contributions
H.S. developed the methods, conducted the analyses and wrote the first draft of the paper. S.C., A.W., N.L. and D.A.T.C. helped methods development. T.B., M.K., J.M.R., I.M.B., S.F., R.J., K.R., S.I., W.V., P.S., C.K., B.T., K.E.-M. and C.B. worked on obtaining data for the analyses. All authors contributed to revising the paper.

## Competing interests
The authors declare no competing interests.
