## [Peer Review File · Nature Communications]

REVIEWER COMMENTS

Reviewer #1 (Remarks to the Author):

Salje et al. developed a analytic framework that utilises the genetic sequences of DENV and their epidemiological information to reconstruct the transmission chains in spatiotemporal dimensions. This study aimed to use the genetic sequences to model the unseen transmission by incorporating human and mosquito data in the transmission model, and thus examine the factors that drive dengue transmission dynamics. They found the infections might have resulted in individuals spending most of daytime in home areas, and majority of transmissions are driven by susceptible individuals that have less mobility (e.g. children). They also found age-dependent mixing of individuals and mosquito distributions are not important to determine the transmission dynamics. Overall, this paper is well-written.

In one of the major results, it found that "the infected individuals spend 96% (95% CI: 88%-100%) of daytime hours within their home cell, compared to 76% (95% CI; 42%- 86%) for susceptible individuals and 42% for adults (95% CI; 41%-43%) (Figure 2A, Figures S3-4)." This finding is interesting – I am amazed by the phylo data and model could make such inference of mobility and infected and uninfected individuals, as the sequence data do not include uninfected individuals. However, I am a bit confused by these estimates, and think that more details would be helpful for understanding. How much % of daytime hours in home cells was estimated for children? What are the proportions of susceptible estimated to be adult and children respectively? Is this proportion supported by epidemiology or other virological observation? Was elderly age group also considered in the adult group? It seems that this age group has a rather different home-cell spending hours than the other adults, but this is probably not reflected by the narrow CI (41-43%) estimated for adult.

This study investigated 726 DENV sequences. Do these sequences form a monophyletic lineage that reflects only local dengue transmission? Do they mix with other foreign sequences, meaning that international importations and exportations occurred? I think such sequence mixing, if not considered, might potentially impact the inference. E.g. If some Thai tree branches in this 726-taxa tree actually split and branched off to other Asian countries, then it means the virus might have gone far from local circulation. This might be particularly true for those long (time) branches. And the long transmission chains in these branches could be mis-estimated to be driven by less mobile susceptible?

Were the ancestral geographic locations of infections estimated/resulted from the model inference? It would be useful to show these locations (for the ancestral nodes) in the trees to better understand the spread of the virus in Thailand.

Was the phylogenetic topological uncertainty considered in the model inference?

Reword - "Current phylogeographic approaches are unable to disentangle the..."

Add another reference - "estimated distributions of vector presence (Figure S2) and the long-term spatio-temporal distribution of serotypes (Figure 1C) Ref#12". The reference #12 cited here provides the estimated vector distribution, but from which source the distribution of serotypes was obtained?

Reviewer #2 (Remarks to the Author):

This study addresses the complex dynamics of dengue transmission in framework that brings new mechanistic insights. Namely, by using sequence differences traditionally modeled with phylogenetic and at best phylogeographic approaches, translating these sequence differences into generation time, and combining this with models of human movement explicit for various demographic and disease states, it allows the testing of various mechanisms potentially underlying dengue transmission dynamics. It is a well-executed study that is innovative in its approach, leverages a unique and powerful data set (in Thailand), and presents impactful results in terms of our understanding of dengue – although it is not new that human movement drives dengue spatial dynamics, this is a much more nuanced and data-driven treatment, and explicitly integrates space and time namely through the footprint of prior immunity. I recommend publication in Nature Communications.

Minor edits/comments:

- Although vector distribution turns out to be unimportant, vector density is an industry standard for predicting risk of dengue transmission and should be treated with a little more explanation, starting here with: “estimated distributions of vector presence (Figure S2)” – for example, consider expanding on source of vector data and presence versus density.

- Clarify/rewrite bold parts: “we estimate that in Bangkok, infected individuals spend 96% (95% CI: 88%-100%) of daytime hours within their home cell, compared to 76% (95% CI; 42%- 86%) for susceptible individuals and 42% for adults (95% CI; 41%-43%) (Figure 2A, Figures S3-4). These differences likely reflect that susceptibility is concentrated in children, who may spend more time near their home than adults. It also suggests that while subclinical DENV infections are common, they typically still result in severe enough symptoms to change daily routine and limit mobility.”

o in this case “These” refers to 76% versus 42%? And “It” refers to 96% versus 76%?

- Too strongly stated/not much of a case for/not biologically significant? Consider revising bold part: “These patterns of differential mobility are also observed at the national scale, with cases spending 96% of the time within their home province (95% CI: 95%-100%), compared to 95% for the susceptible population (95% CI: 92%- 100%) and 87% for adults (95% CI: 86%-88%) (Figure 2B).”

- What is the justification for considering “recent” hetero- versus homotypic circulation as 1-2 years? As stated here: “with transmission more likely to occur in places that have seen increases of other (heterotypic) serotypes circulating in the previous two years and less likely to occur in places with increased cases of the same (homotypic) serotype within the same timeframe (Figure 2D-E, Figures S3-4).” Do not these immune responses persist over longer time frames, and is there a better way to measure serological landscape (e.g. with sero-surveys)?

- I am unfamiliar with the phrase “held out” and suggest it be replaced with “unsampled,” if that is what is meant.

- Not sure I completely understand the significance of these 2 statements: “Overall, within Bangkok, we find that 34% of infections occur outside the 1km² home grid cell of an infected individual (95%CI: 26%-43%). This is compared to 4% of individual case mobility being outside cells (Figure 3C).”

Signed,
Shannon Bennett

Reviewer #3 (Remarks to the Author):

Very nice paper by Salje et al to fuse geographic and genetic data to make inferences on the epidemiology of dengue in Thailand. Importantly, this is not just a methods paper, although the methodology they outline in this paper could be deployed for other pathogen systems, but also provides important insights on dengue in Thailand, which despite considerable amounts of study retains hidden features. I do however have some queries that the authors might address to improve the paper further.

The paper does not have page or line numbers so I'll copy and paste bits of the pdf here to help identify where the comment refers. It would be good practice to require page and line numbers for future submissions to reduce the work of the reviewers.

My main concern is the finding that cases spend almost 100% of their time in the home cell (in BKK). I can believe this for provinces (since 'home' is the province, not the 1km cell) but struggle to see how it would apply in the capital. Dengue infections fall into several categories: symptomatic infections that are diagnosed by the healthcare system and notified; symptomatic infections that are not diagnosed; and asymptomatic infections that are not diagnosed (assuming no active case finding). The jury is out on the ratio of subclinical symptomatic and asymptomatic infections, but the authors' finding is suggestive of all of these falling into the symptomatic category, since asymptomatic infections would presumably follow the movement patterns of the susceptible population, who seem to spend about 80% of their time in the home cell. To be honest I don't believe this finding and would want to have some supporting evidence that is not model-based to justify it. Bear in mind also that people who are symptomatic are infectious to mosquitoes prior to onset of symptoms.

1. will be applicable across pathogens: I'd avoid overselling the work. The inference I got from this was that ALL pathogens can now be analysed with this approach but for instance can food-borne or nosocomial bacteria be analysed with this approach? Probably these would require such extensive modifications that it'd be a new method.

2. which in the case of DENV will only ever represent <1% of all infections: quite a confusing sentence. I can't tell if you mean that cases of dengue can maximally be 1/100 of dengue infections or whether you mean that sequenced dengue cases are <1/100 of dengue infections. If the former, then it's not true: eg Singapore detects 10-20% of its infections. If the latter, then surely it depends on the amount of sequencing being done, which may in future increase. Eg for TB some places are now aiming to sequence all isolates which would have been inconceivable years ago.

3. Timing of data sources. The dengue data cover 1995 to 2012 (it is surprising and a little saddening that more up-to-date data were not available) but the telco data are from 2017. Population data were from what period? Can the authors note the limitation and speculate on the impacts of having mismatched data sources?

4. I don't see the point of the comparison vs polymod. For a start, polymod is for EU countries. Thailand has a different population structure, income level, and cultural practices from places like Germany and the UK. But more importantly, polymod is a measure of social contacts---ie it informs transmission of influenza, SARS-CoV-2 etc---not of shared geographic contacts, as with dengue. Thirdly, polymod speaks of potential infection, but your approach (unless I misunderstand it) is of actual infections, which is affected by age-mediated immunity. To be honest, the value of this part of the work seems negligible, so given that the data don't correspond to Thailand anyway I would just remove it.

5. The submodel on transmission I found a bit confusing. In the main part of the paper, it is said that "We find that if a virus is introduced into a randomly selected province, it is on average 4.3 times as likely to have travelled to Bangkok after a single transmission generation compared to anywhere else". I think the phrasing is incorrect: if the virus is introduced to Yala, it is not going to be 4 times more likely to be exported to BKK than anywhere else, it is more likely to be exported to Pattani, surely? Additionally, when the authors write about it being 11 times more likely to be in BKK after 20 generations, there are several conflicting ways to interpret that: that the probability

of having 1+ surviving descendant in BKK is 11 times more than the probability of having 1+ descendant than in province X, that the probability of having 1+ surviving descendant in BKK is 11 times more than the probability of having 1+ descendant than in any other province, that the probability of a given descendant being in BKK is 11 times more than the probability of a given descendant being in province X, etc.

6. Figure 1: possible to have larger phylogenetic trees in the supplement? I struggle to get anything from these tiny charts.

7. Figure 2: First, what are these boxplots supposed to represent? Assuming it's an uncertainty interval, panel F is a bit of a concern. Why is the uncertainty interval for the mosquito exponent strictly between -5 and 5? If this were a Bayesian analysis I would have thought this was the effect of a uniform prior with overly narrow ranges. Given that the authors are doing ML estimation, what would be the cause of this? Are they only searching over the range [-5,5]? I think this should be reassessed. More fundamentally, why can the model not estimate the impact of mosquitoes on transmission? (on a minor point, I really struggle to see the black lines on these dark greens and blues. Please consider lightening the boxes or making the central line white rather than black. I struggle also to distinguish the points on panel C: perhaps one symbol could be hollow if you need to use these tones? Also the y ticks on panel C are haphazardly placed)

8. We used a truncated log-normal distribution with a mean of 5.6 days: please clarify, the mean of 5.6 is before truncating or after truncating?

9. We obtained a mean generation time of 18.2 days and a variance of 37.2 days: sorry to be anal but the variance should be 37.2 square-days. If the notion of square-time sits as uncomfortably with you as it does me, then you should report the standard deviation, which thankfully has real-world units.

10. The number of susceptible individuals living in a location: possible to show this in a supplemental figure?

11. The methods has a lot of subordinate clauses with capital "Where"s at the beginning, looking like sentences. Please do check the grammar.

12. To calculate the probability of the home location being within location k after G transmission generations, we can use matrix multiplication that integrates over all possible pathways connecting two locations: the equation that follows has some non-standard notation that looks like R code?

13. In order to incorporate uncertainty, we use a bootstrapping approach where we randomly sample all the available sequences with replacement over 100 iterations and recalculate the maximum likelihood estimate for each parameter each time. The 95% confidence intervals are then taken from the 2.5 and 97.5 percentiles of the resulting distribution: If you only do 100 resamples then you're effectively taking the mean of the 2nd and 3rd lowest to get the 2.5th percentile. The problem with this is that the number of resamples is so small there must be a lot of uncertainty on the 2nd and 3rd values. Given that this uncertainty is under your control (i.e. not due to the size of the dataset) it feels a bit wrong to stop at just 100 iterations. If really Cambridge, Florida, Pasteur etc don't have enough computing power to get up to 1000 iterations, then perhaps you might consider using a normal approximation instead which is presumably less prone to stochastic error than selecting values in the tails?

14. We generate daughter infections from the index using a random draw from a Poisson distribution: shouldn't the distribution of daughters have extra-poisson variability?

15. To characterize the susceptibility, we assume that the probability of being susceptible is equal to $\frac{1}{N_{age a}}$ where $N_{age a}$ is the number of people of age a: why would the probability of my being susceptible depend linearly on how many people live in my area?

16. Figure S2: I find it hard to believe that the average person in any province of Thailand has a less than 50% chance of living near *Ae aegypti*. Are these weighted based on area or population? Probably it should be the latter?

17. Figure S10: missing y axis label for panels A and E. Consider putting $R_{eff}=1.6$ top in the legends to follow the ordering of the curves on the plot.

REVIEWER COMMENTS

Reviewer #1 (Remarks to the Author):

Comment 1.1. *Salje et al. developed a analytic framework that utilises the genetic sequences of DENV and their epidemiological information to reconstruct the transmission chains in spatiotemporal dimensions. This study aimed to use the genetic sequences to model the unseen transmission by incorporating human and mosquito data in the transmission model, and thus examine the factors that drive dengue transmission dynamics. They found the infections might have resulted in individuals spending most of daytime in home areas, and majority of transmissions are driven by susceptible individuals that have less mobility (e.g. children). They also found age-dependent mixing of individuals and mosquito distributions are not important to determine the transmission dynamics. Overall, this paper is well-written.*

Response: We thank the reviewer for their thoughtful feedback.

Comment 1.2. *In one of the major results, it found that “the infected individuals spend 96% (95% CI: 88%-100%) of daytime hours within their home cell, compared to 76% (95% CI; 42%- 86%) for susceptible individuals and 42% for adults (95% CI; 41%-43%) (Figure 2A, Figures S3-4).” This finding is interesting – I am amazed by the phylo data and model could make such inference of mobility and infected and uninfected individuals, as the sequence data do not include uninfected individuals. However, I am a bit confused by these estimates, and think that more details would be helpful for understanding. How much % of daytime hours in home cells was estimated for children? What are the proportions of susceptible estimated to be adult and children respectively? Is this proportion supported by epidemiology or other virological observation? Was elderly age group also considered in the adult group? It seems that this age group has a rather different home-cell spending hours than the other adults, but this is probably not reflected by the narrow CI (41-43%) estimated for adults.*

Response: The adult mobility is taken directly from mobile phone data. DTAC (a major mobile phone operator in Thailand) have provided summary information on the cell masts used by individuals throughout the country. The mobility estimates are the probability that a call is made in location j for an individual that lives in location i over all hours of the day. It has been estimated that 67% of calls are between 8am and 7pm (Aledavood et al. 2015; Liu et al. 2018) - we therefore feel that it is reasonable that the estimates of human mobility from CDR will be largely representative of daytime mobility, however, we agree that daytime mobility may be slightly different than this summary measure. We have clarified this in the revised manuscript.

The CDR information is provided at an aggregate level only - all we know is that individuals are all adults. We therefore cannot explore differences by age. Instead, we use this CDR as a base for estimating mobility for susceptible individuals and infected individuals. The mobility for infected and susceptible individuals is estimated by our model by adjusting the (adult) mobility from the

CDR data by allowing for different amounts of time spent within the home cell through separate estimated parameters (one parameter for the susceptible population and a separate one for infected individuals). The movement to each non-home location is scaled up or down to ensure that total mobility from each cell sums to 1 (ie all mobility is accounted for). Therefore we are estimating mobility matrices for 'susceptible people' and 'infected people'. However, as dengue has circulated for a long time in Thailand, we know that most adults are immune to infection and therefore susceptible individuals are mainly children.

To further validate our findings of different patterns of mobility between adults (measured from CDR) and susceptible individuals (estimated from the model) - we have now incorporated data from a travel survey (the SMILI project) that specifically asked randomly selected individuals of all ages about their daily movement patterns (see figure below). We find that, consistent with our findings, child participants in the SMILI study reported spending more time around their home as compared to adults. Once we incorporate the estimated susceptibility to dengue by age, we find that, on average, the findings from the SMILI project indicate that susceptible individuals were around 50% more likely to have not travelled further than 1km from their home than adults (see figure below, now Figure S10). This is very similar to our model estimates of the difference in human mobility between susceptible individuals and adults and therefore provide a strong validation of our approach.

Specific changes to manuscript: We now include :”Dengue transmission is more likely to occur during daylight hours, due to the feeding behaviour of Aedes mosquitoes. Therefore, it would be optimal to use CDR data from daylight hours only. However, as is often the case, our CDR data represents an aggregate from all hours of the day. Nevertheless, as the majority of cell phone calls are made during daylight hours (it has been estimated that 67% of calls are between 8am and 7pm) (Aledavood et al. 2015; Liu et al. 2018) - it is reasonable that this estimate is largely representative of daytime mobility. “

In addition, we now include: “As dengue susceptibility is concentrated in children (Figure S10), our findings of reduced mobility in susceptible individuals suggests that children are less likely to travel far from their home than adults. To explore the consistency of this finding with observed differences in mobility by age, we use data from a separate study from Thailand that asked individuals (N=2011) of all ages about their daily travel. Consistent with our findings of reduced mobility in susceptible individuals, we find that there is a strong relationship between age and reporting having stayed within 1km of their home in the prior week (Figure S10). Incorporating the probability of being susceptible by age suggests that susceptible individuals are 1.5 (95%CI: 1.2-1.9) times as likely to report staying within 1km of their home in the last seven days, consistent with the 1.8 (95%CI: 1.3-2.2) times difference estimated by our model (Figure S10).”

Finally, we say: “[our findings] show that on average, infected individuals are more likely to stay in and around their home. This suggests that some subclinical DENV infections may still result in severe enough symptoms to change daily routine and limit mobility. Further observational studies are needed to understand how movement changes across the spectrum of disease severity(Perkins et al. 2016). “

Human mobility data from SMILI project (N=2011). **(A)** Proportion of individuals that reported having not travelled more than 1km within the last 7 days by 5 year age group with 95% confidence intervals. **(B)** Proportion of the population by five-year age group that is susceptible to infection (completely naive or monotypically immune) assuming a fixed force of infection of 0.04 per serotype/year. **(C)** Estimated average proportion of the susceptible population and adult population that has not travelled more than 1km in the last seven days with 95% confidence intervals and interquartile range. **(D)** Relative risk of having not travelled more than 1km in the last seven days comparing the susceptible population with adults (blue) and the relative risk of staying within a grid cell for susceptible individuals versus adults as inferred by the Bangkok model (green).

Comment 1.3. *This study investigated 726 DENV sequences. Do these sequences form a monophyletic lineage that reflects only local dengue transmission? Do they mix with other foreign sequences, meaning that international importations and exportations occurred? I think such sequence mixing, if not considered, might potentially impact the inference. E.g. If some Thai tree branches in this 726-taxa tree actually split and branched off to other Asian countries, then it means the virus might have gone far from local circulation. This might be particularly true for those long (time) branches. And the long transmission chains in these branches could be mis-estimated to be driven by less mobile susceptible?*

Response: The four serotypes of dengue have circulated in Thailand for decades (at least since the 1950s). We have previously shown (Salje et al., Science 2017) that the epidemic in Thailand is self-sustaining with very few foreign introductions - essentially you never get sequences from one of the neighbouring countries that have an MRCA with a Thai virus within the previous year. In this project, we have developed a method that fits models to pairs of sequences that are separated by a year or less (equivalent to around 20 transmission generations) - with sensitivity analyses that demonstrate that it is robust to this cut off (Figure S9). The rationale is that after 20 transmission generations, much of the spatial structure will be lost. We therefore do not require a single monophyletic lineage across all the sequences.

However, we agree that there is an inherent assumption that our spatial area of analysis does constitute the only locations people (and the virus) can travel to. We have clarified this in the revised manuscript.

Specific changes to manuscript: We now include “As the sum of movements to the destinations in the matrix is equal to 1, we are assuming that the spatial unit of analysis contains all possible mobility (of the virus and people). It has previously been shown that the dengue epidemic in Thailand is self-sustaining with few external introductions (Salje et al. 2017). Applications of this approach to small spatial units should consider that some mobility may be missed. “

Comment 1.4. *Were the ancestral geographic locations of infections estimated/resulted from the model inference? It would be useful to show these locations (for the ancestral nodes) in the trees to better understand the spread of the virus in Thailand.*

Response: We did not directly reconstruct ancestral locations of lineages. Instead we are integrating over all possibilities using the viral movement matrices. We also limit the analyses to pairs of sequences that are separated by under 20 transmission generations (equivalent to around a year of sequential transmissions). We take this approach as the spatial signal will get progressively lost with more and more transmission generations. We obtain essentially identical results when we extend the cutoff to 40 generations in a sensitivity analysis (Figure S9). Therefore most ancestral nodes in a tree would be prior to this cut-off and not included in the analysis. It would theoretically be possible to take the final fitted model and go back and identify the likeliest

location at each transmission generation - however, this would represent a substantial amount of work and we do not feel that it would add much to the manuscript.

Comment 1.5. *Was the phylogenetic topological uncertainty considered in the model inference?*

Response: Yes - we integrate over 100 randomly selected posterior trees for each serotype. We clarify this in the revised manuscript.

Specific changes to manuscript: The paper now includes “We fit our models in a maximum likelihood framework that incorporates uncertainty from the evolutionary processes, including topological uncertainty in the phylogenetic trees, uncertainty in the generation time distribution, and sampling uncertainty using a bootstrap approach.”

Comment 1.6. *Reword - “Current phylogeographic approaches are unable to disentangle the...”*

Response: We have adjusted this sentence to make it clear that many existing phylogeographic approaches do not consider intervening transmission events in unsampled locations.

Specific changes to manuscript: This sentence now reads “Many existing phylogeographic approaches will infer a viral flow between observed locations without consideration that transmission events that link two observed sequences will be unobserved and often in unsampled locations.”

Comment 1.7. *Add another reference - “estimated distributions of vector presence (Figure S2) and the long-term spatio-temporal distribution of serotypes (Figure 1C) Ref#12”. The reference #12 cited here provides the estimated vector distribution, but from which source the distribution of serotypes was obtained?*

Specific changes to manuscript: We now include a reference to Salje et al. Science 2017 that presents underlying data on serotype distribution across Bangkok and Thailand.

Reviewer #2 (Remarks to the Author):

Comment 2.1. *This study addresses the complex dynamics of dengue transmission in framework that brings new mechanistic insights. Namely, by using sequence differences traditionally modeled with phylogenetic and at best phylogeographic approaches, translating these sequence differences into generation time, and combining this with models of human movement explicit for various demographic and disease states, it allows the testing of various mechanisms potentially underlying dengue transmission dynamics. It is a well-executed study that is innovative in its approach, leverages a unique and powerful data set (in Thailand), and presents impactful results in terms of our understanding of dengue – although it is not new that human movement drives dengue spatial dynamics, this is a much more nuanced and data-driven treatment, and explicitly integrates space and time namely through the footprint of prior immunity. I recommend publication in Nature Communications.*

Response: We thank the reviewer for this generous feedback.

Minor edits/comments:

Comment 2.2. - *Although vector distribution turns out to be unimportant, vector density is an industry standard for predicting risk of dengue transmission and should be treated with a little more explanation, starting here with: “estimated distributions of vector presence (Figure S2)” – for example, consider expanding on source of vector data and presence versus density.*

Response: We agree that this was not clearly set out. The vector data comes from modelled estimates of *Aedes aegypti* occurrence. The relationship between this value and vector density remains unclear. We have clarified this in the revised manuscript.

Specific changes to manuscript: In the main text, we have changed the wording to say “modelled estimates of the probability of *Aedes aegypti* occurrence”.

In the methods, we have more detail on the source of the data. “We used previously published estimates of the probability of *Aedes aegypti* presence for 5km x 5km grid cells around the globe¹⁷. These estimates were generated by incorporating information on temperature, rainfall, vegetation indices from satellite imagery and fitting models to a large dataset of *Aedes* occurrence records. The fitted models were then used to predict elsewhere.”.

We also clarify that the relationship between probability of occurrence and density remains unclear: “This does not rule out a role for the vector in characterizing heterogeneity in risk. In particular, the relationship between modelled probabilities of occurrence that we have used and vector density remains unclear. “

Comment 2.3. - Clarify/rewrite **starred** words: “we estimate that in Bangkok, infected individuals spend 96% (95% CI: 88%-100%) of daytime hours within their home cell, compared to 76% (95% CI; 42%- 86%) for susceptible individuals and 42% for adults (95% CI; 41%-43%) (Figure 2A, Figures S3-4). **These** differences likely reflect that susceptibility is concentrated in children, who may spend more time near their home than adults. **It** also suggests that while subclinical DENV infections are common, they typically still result in severe enough symptoms to change daily routine and limit mobility.”

o in this case **These** refers to 76% versus 42%? And **It** refers to 96% versus 76%?

Response: Thank you for highlighting this confusing text. It has been rewritten.

Specific changes to manuscript: This section now reads:

“Using our framework, we estimate that in Bangkok, susceptible individuals spend 76% (95% CI; 57%-95%) of their time within their home cell as compared to 42% for adults (95% CI; 41%-43%) (Figure 2A, Figures S7-9). As dengue susceptibility is concentrated in children (Figure S10), our findings of reduced mobility in susceptible individuals suggests that children are less likely to travel far from their home than adults. “

...Using our model, we find that infected individuals in Bangkok are even less mobile than susceptible individuals with 96% (95% CI: 87%-100%) of infected individuals’ time being within their home cell (Figure 2A). Importantly, these estimates of differential mobility hold for the intervening unseen transmission events, as well as the observed cases in the phylogeny. This shows that on average, infected individuals are more likely to stay in and around their home. This suggests that some subclinical DENV infections may still result in severe enough symptoms to change daily routine and limit mobility. Further observational studies are needed to understand how movement changes across the spectrum of disease severity¹³. We also observed similar differences in mobility patterns at the national scale, with cases spending 96% of the time within their home province (95% CI: 86%-100%), compared to 95% for the susceptible population (95% CI: 89%-100%) and 87% for adults (95% CI: 86%-88%) (Figure 2B).”

Comment 2.4. - Too strongly stated/not much of a case for/not biologically significant? Consider revising **starred** part: “These **patterns of differential mobility** are also observed at the national scale, with cases spending 96% of the time within their home province (95% CI: 95%-100%), compared to 95% for the susceptible population (95% CI: 92%- 100%) and 87% for adults (95% CI: 86%-88%) (Figure 2B)

Response: This has been rewritten.

Specific changes to manuscript: This section now reads: “We also observed similar differences in mobility patterns at the national scale, ...”

Comment 2.5. - *What is the justification for considering “recent” hetero- versus homotypic circulation as 1-2 years? As stated here: “with transmission more likely to occur in places that have seen increases of other (heterotypic) serotypes circulating in the previous two years and less likely to occur in places with increased cases of the same (homotypic) serotype within the same timeframe (Figure 2D-E, Figures S3-4).” Do not these immune responses persist over longer time frames, and is there a better way to measure serological landscape (e.g. with sero-surveys)?*

Response: We agree this required some additional justification. National seroprevalence surveys would be a fantastic resource that could absolutely help with characterising the infection risk landscape, however these are not available - in particular they would need to be done on a regular basis to capture the changing patterns of risk, be serotype specific (and therefore require e.g., PRNTs) and cover all areas of the country. Instead, we use serotype-specific incidence as a proxy for state of local immunity. We have previously found that local immunity in Thailand appears to be spatially correlated for time frames of under 2 years (Salje et al., PNAS 2012), we have therefore used that as a basis for our choice of a 2 year window. We have included this justification in the revised manuscript.

Specific changes to manuscript: We now include in the relevant section in the methods: “We choose a window of 2 years to define recent immunity as serotype-specific incidence has previously been shown to be spatially correlated over this time range, presumably due to serotype-specific local herd immunity(Salje et al. 2012). “

We also include: “More direct measures of local immunity through population-representative seroprevalence studies may provide a more nuanced picture of the role of immunity in patterns of spread(Salje et al. 2019; Metcalf et al. 2016). “

Comment 2.6. - *I am unfamiliar with the phrase “held out” and suggest it be replaced with “unsampled,” if that is what is meant.*

Response: We define ‘held out’ in the revised manuscript.

Specific changes to manuscript: This section now reads “In order to assess the performance of our model, we repeatedly refit the model on data where we remove all sequences from a subset of locations (held out locations). We find our model is able to accurately estimate the probability of observing viruses in the held out locations, both within Bangkok and at the nationwide scale...”

Comment 2.7. - *Not sure I completely understand the significance of these 2 statements: “Overall, within Bangkok, we find that 34% of infections occur outside the 1km² home grid cell of an infected individual (95%CI: 26%-43%). This is compared to 4% of individual case mobility being outside cells (Figure 3C).”*

Response: The point here is that while cases don't move a lot, the distance between the homes of two sequential cases in a transmission chain can nevertheless be far apart due to the mobility of the susceptible population. We have attempted to clarify this in the revised manuscript.

Specific changes to manuscript: This section now reads: “Overall, within Bangkok, we find that 34% of infections occur outside the 1km² home grid cell of an infected individual (95%CI: 26%-43%). This is despite infected individuals spending only an average of 4% of their time outside their home cells, highlighting the importance of considering mobility in both infected and susceptible populations when considering viral spread (Figure 3C).”

Reviewer #3 (Remarks to the Author):

Comment 3.1. *Very nice paper by Salje et al to fuse geographic and genetic data to make inferences on the epidemiology of dengue in Thailand. Importantly, this is not just a methods paper, although the methodology they outline in this paper could be deployed for other pathogen systems, but also provides important insights on dengue in Thailand, which despite considerable amounts of study retains hidden features. I do however have some queries that the authors might address to improve the paper further.*

Response: We thank the reviewer for these kind words.

Comment 3.2. *The paper does not have page or line numbers so I'll copy and paste bits of the pdf here to help identify where the comment refers. It would be good practice to require page and line numbers for future submissions to reduce the work of the reviewers.*

Response: We agree and have included them in the revised document.

Comment 3.3. *My main concern is the finding that cases spend almost 100% of their time in the home cell (in BKK). I can believe this for provinces (since 'home' is the province, not the 1km cell)*

but struggle to see how it would apply in the capital. Dengue infections fall into several categories: symptomatic infections that are diagnosed by the healthcare system and notified; symptomatic infections that are not diagnosed; and asymptomatic infections that are not diagnosed (assuming no active case finding). The jury is out on the ratio of subclinical symptomatic and asymptomatic infections, but the authors' finding is suggestive of all of these falling into the symptomatic category, since asymptomatic infections would presumably follow the movement patterns of the susceptible population, who seem to spend about 80% of their time in the home cell. To be honest I don't believe this finding and would want to have some supporting evidence that is not model-based to justify it. Bear in mind also that people who are symptomatic are infectious to mosquitoes prior to onset of symptoms.

Response: We thank the reviewer for their comment. We initially shared the reviewer's slight scepticism at this result, however, we have performed extensive simulations and found that we can indeed recover differential mobility between infected and susceptible people (Figure S13). In addition, we now include data from a separate study from Thailand, which shows differential mobility between children and adults - which is consistent with our findings of differences in mobility between susceptible individuals and adults, again suggesting our model is able to identify such mobility features in the data (details below). We agree that it would also be useful to also have observational studies that explore mobility of infected individuals, especially those with few symptoms, however, we are not aware of such studies. In the revised manuscript, we highlight that our findings represent population-average results (ie it averages over all infections and we do not know the symptom status of intermediary infections) and that further work is needed to understand how individual-level mobility patterns change following infection.

We now include data from mobility questionnaires in Thailand (SMILI project) that show that children move less than adults. The SMILI project randomly selected individuals from around Thailand of all ages and asked them about their daily mobility patterns. We find that, consistent with our findings, children spend more time around their home as compared to adults. Once we incorporate the estimated susceptibility to dengue by age, we find that, on average, the findings from the SMILI project indicate that susceptible individuals were around 50% more likely to have not travelled further than 1km from their home than adults (see figure below, now Figure S10). This is very similar to our model estimates of the difference in human mobility between susceptible individuals and adults and therefore provide a strong validation of our approach.

Specific changes to manuscript: We have now included an analysis of age-specific mobility from questionnaire data. In the main text we say: "Using our framework, we estimate that in Bangkok, susceptible individuals spend 76% (95% CI; 57%-95%) of their time within their home cell as compared to 42% for adults (95% CI; 41%-43%) (Figure 2A, Figures S7-9). As dengue susceptibility is concentrated in children (Figure S10), our findings of reduced mobility in susceptible individuals suggests that children are less likely to travel far from their home than adults. To explore the consistency of this finding with observed differences in mobility by age, we use data from a separate study from Thailand that asked individuals (N=2011) of all ages about their daily travel. Consistent with our findings of reduced mobility in susceptible individuals, we

find that there is a strong relationship between age and reporting having stayed within 1km of their home in the prior week (Figure S10). Incorporating the probability of being susceptible by age suggests that susceptible individuals are 1.5 (95%CI: 1.2-1.9) times as likely to report staying within 1km of their home in the last seven days, consistent with the 1.8 (95%CI: 1.3-2.2) times difference estimated by our model (Figure S10).”

Using our model, we find that infected individuals in Bangkok are even less mobile than susceptible individuals with 96% (95% CI: 87%-100%) of infected individuals’ time being within their home cell (Figure 2A). Importantly, these estimates of differential mobility hold for the intervening unseen transmission events, as well as the observed cases in the phylogeny. This shows that on average, infected individuals are more likely to stay in and around their home. This suggests that some subclinical DENV infections may still result in severe enough symptoms to change daily routine and limit mobility. Further observational studies are needed to understand how movement changes across the spectrum of disease severity.”

Human mobility data from SMILI project (N=2011). (A) Proportion of individuals that reported having not travelled more than 1km within the last 7 days by 5 year age group with 95% confidence intervals. (B) Proportion of the population by five-year age group that is susceptible to infection (completely naive or monotypically immune) assuming a fixed force of infection of 0.04 per serotype/year. (C) Estimated average proportion of the susceptible population and adult population

that has not travelled more than 1km in the last seven days with 95% confidence intervals and interquartile range. **(D)** Relative risk of having not travelled more than 1km in the last seven days comparing the susceptible population with adults (blue) and the relative risk of staying within a grid cell for susceptible individuals versus adults as inferred by the Bangkok model (green).

Comment 3.4. *1. will be applicable across pathogens: I'd avoid overselling the work. The inference I got from this was that ALL pathogens can now be analysed with this approach but for instance can food-borne or nosocomial bacteria be analysed with this approach? Probably these would require such extensive modifications that it'd be a new method.*

Response: We agree with the reviewer and have changed this wording.

Specific changes to manuscript: This now reads: "...will be applicable to other pathogens:. In the conclusion we also state: "While we have used this framework for DENV, it is applicable to other communicable pathogens where there exists a time-resolved phylogeny, the generation time distribution is known and is relatively short (days or weeks) and there exists spatial information or other discrete traits."

Comment 3.5. *which in the case of DENV will only ever represent <1% of all infections: quite a confusing sentence. I can't tell if you mean that cases of dengue can maximally be 1/100 of dengue infections or whether you mean that sequenced dengue cases are <1/100 of dengue infections. If the former, then it's not true: eg Singapore detects 10-20% of its infections. If the latter, then surely it depends on the amount of sequencing being done, which may in future increase. Eg for TB some places are now aiming to sequence all isolates which would have been inconceivable years ago.*

Response: This statement was meant to refer to the proportion of infections that are sequenced. We have clarified this and made sure we refer to current efforts.

Specific changes to manuscript: This now reads "In addition, fewer than 1% of dengue infections will currently be sequenced from any one country. "

Comment 3.6. *Timing of data sources. The dengue data cover 1995 to 2012 (it is surprising and a little saddening that more upto-date data were not available) but the telco data are from 2017. Population data were from what period? Can the authors note the limitation and speculate on the impacts of having mismatched data sources?*

Response: We agree that this is an important consideration. The population data are from 2010. The Thai population is relatively stable (rising from 60 million to 64 million between 1995 and 2012). We have noted the limitation of mismatched timing of data sources in the revised manuscript.

Specific changes to manuscript: We now include: “We also note that the mobile phone data was collected after our study period (2017 vs 1995-2012). Human mobility may have changed and could help explain some of the differences in mobility between the fitted models and that implied from the mobile phone data.”

We also include: “...we used population size estimates from the 2010 national census. We note that the Thai population has been relatively stable over the study period (rising from 60 million to 64 million between 1995 and 2012). “

Comment 3.5. *I don't see the point of the comparison vs polymod. For a start, polymod is for EU countries. Thailand has a different population structure, income level, and cultural practices from places like Germany and the UK. But more importantly, polymod is a measure of social contacts--ie it informs transmission of influenza, SARS-CoV-2 etc---not of shared geographic contacts, as with dengue. Thirdly, polymod speaks of potential infection, but your approach (unless I misunderstand it) is of actual infections, which is affected by age-mediated immunity. To be honest, the value of this part of the work seems negligible, so given that the data don't correspond to Thailand anyway I would just remove it.*

Response: We had gone back and forth on whether this was a useful addition. We were using it as a reference point to what we would expect if there was age-assortative infection. However, we agree that it is confusing. We did specifically take account of immunity in the model - this is actually one of the key points - if you ignore immunity, it does look like mixing by age but once you account for immunity, that goes away. On balance, we agree with the reviewers and have removed the Polymod aspect.

Specific changes to manuscript: We have removed the POLYMOD results.

Comment 3.6. *The submodel on transmission I found a bit confusing. In the main part of the paper, it is said that “We find that if a virus is introduced into a randomly selected province, it is on average 4.3 times as likely to have travelled to Bangkok after a single transmission generation compared to anywhere else”. I think the phrasing is incorrect: if the virus is introduced to Yala, it is not going to be 4 times more likely to be exported to BKK than anywhere else, it is more likely to be exported to Pattani, surely? Additionally, when the authors write about it being 11 times more likely to be in BKK after 20 generations, there are several conflicting ways to interpret that: that the probability of having 1+ surviving descendant in BKK is 11 times more than the probability of having 1+ descendant than in province X, that the probability of having 1+ surviving descendant in BKK is 11 times more than the probability of having 1+ descendant than in any other province,*

that the probability of a given descendant being in BKK is 11 times more than the probability of a given descendant being in province X, etc.

Response: We agree it was confusing as written. The results are an average across repeated simulations where the initial infection is placed in a randomly selected province. Therefore - the results are an average over all possible places of introduction. We have clarified these sentences in the revised manuscript.

Specific changes to manuscript: This section now reads: “We use a simulation approach that introduces viruses into randomly selected provinces and use the fitted mobility matrices to see where transmission occurs over 20 transmission generations. Averaging over repeated simulations, we find that the virus is 4.3 (95%CI: 2.4-7.0) times as likely to have travelled to Bangkok after a single transmission generation as compared to a randomly selected province. After 20 transmission generations (equivalent to approximately one year of sequential transmissions) we find the virus is 11.4 (95%CI: 6.3-19.4) times as likely to have infected at least one individual in the capital as compared to at least one individual living in a randomly selected province (Figures 3A, S8). “

Comment 3.7. *Figure 1: possible to have larger phylogenetic trees in the supplement? I struggle to get anything from these tiny charts.*

Specific changes to manuscript: Larger phylogenetic trees are now included in the supplementary materials.

Comment 3.8. *Figure 2: First, what are these boxplots supposed to represent? Assuming it's an uncertainty interval, panel F is a bit of a concern. Why is the uncertainty interval for the mosquito exponent strictly between -5 and 5? If this were a Bayesian analysis I would have thought this was the effect of a uniform prior with overly narrow ranges. Given that the authors are doing ML estimation, what would be the cause of this? Are they only searching over the range [-5,5]? I think this should be reassessed. More fundamentally, why can the model not estimate the impact of mosquitoes on transmission? (on a minor point, I really struggle to see the black lines on these dark greens and blues. Please consider lightening the boxes or making the central line white rather than black. I struggle also to distinguish the points on panel C: perhaps one symbol could be hollow if you need to use these tones? Also the y ticks on panel C are haphazardly placed)*

Response: Thank you for highlighting this. The boxplots represent 95% confidence intervals as well as interquartile ranges. The mosquito exponents were not strictly between -5 and 5. It should have read <-5 and >5 (essentially there is enormous uncertainty). In simulated data, we were able to estimate an impact of mosquito occurrence where there was one (Figure S13). We have altered the figures to make them easier to read too.

Specific changes to manuscript: We have altered the figures as requested and included descriptions of the boxplots in the captions.

Comment 3.9. We used a truncated log-normal distribution with a mean of 5.6 days: please clarify, the mean of 5.6 is before truncating or after truncating?

Specific changes to manuscript: We have clarified that this mean is prior to truncation.

Comment 3.10. We obtained a mean generation time of 18.2 days and a variance of 37.2 days: sorry to be anal but the variance should be 37.2 square-days. If the notion of square-time sits as uncomfortably with you as it does me, then you should report the standard deviation, which thankfully has real-world units.

Specific changes to manuscript: We agree and now report the standard deviation.

Comment 3.11. The number of susceptible individuals living in a location: possible to show this in a supplemental figure?

Specific changes to manuscript: In the revised manuscript, we include figures setting out the number of susceptible individuals across the two spatial scales.

Comment 3.12. The methods has a lot of subordinate clauses with capital "Where"s at the beginning, looking like sentences. Please do check the grammar.

Specific changes to manuscript: These have all been corrected.

Comment 3.13. To calculate the probability of the home location being within location k after G transmission generations, we can use matrix multiplication that integrates over all possible pathways connecting two locations: the equation that follows has some non-standard notation that looks like R code?

Specific changes to manuscript: We have removed this syntax.

Comment 3.14. *In order to incorporate uncertainty, we use a bootstrapping approach where we randomly sample all the available sequences with replacement over 100 iterations and recalculate the maximum likelihood estimate for each parameter each time. The 95% confidence intervals are then taken from the 2.5 and 97.5 percentiles of the resulting distribution: If you only do 100 resamples then you're effectively taking the mean of the 2nd and 3rd lowest to get the 2.5th percentile. The problem with this is that the number of resamples is so small there must be a lot of uncertainty on the 2nd and 3rd values. Given that this uncertainty is under your control (i.e. not due to the size of the dataset) it feels a bit wrong to stop at just 100 iterations. If really Cambridge, Florida, Pasteur etc don't have enough computing power to get up to 1000 iterations, then perhaps you might consider using a normal approximation instead which is presumably less prone to stochastic error than selecting values in the tails?*

Specific changes to manuscript: In the revised manuscript we use a normal approximation as kindly suggested by the reviewer. The results are largely unchanged.

Comment 3.15. *We generate daughter infections from the index using a random draw from a Poisson distribution: shouldn't the distribution of daughters have extra-poisson variability?*

Response: We explore using an over-dispersed distribution through a negative binomial distribution and obtain consistent results.

Specific changes to manuscript: We include a new figure (Figure S17) with the results of the over-dispersed simulations. We also include in the text: "We observe consistent patterns across different effective reproductive numbers and for overdispersed transmission (Figures S16-17)"

Comment 3.16. *To characterize the susceptibility, we assume that the probability of being susceptible is equal to $\frac{1}{N_{age a}}$ where $N_{age a}$ is the number of people of age a : why would the probability of my being susceptible depend linearly on how many people live in my area?*

Response: This was a typo - this should have read that the number of susceptible people is equal to....

Specific changes to manuscript: This section now reads: "To characterize the susceptibility, we assume that the number of susceptible people of age a is equal to ..."

Comment 3.17. *Figure S2: I find it hard to believe that the average person in any province of Thailand has a less than 50% chance of living near Ae aegypti. Are these weighted based on area or population? Probably it should be the latter?*

Response: These estimates come directly from the modelled estimates of *Aedes aegypti* probabilities of occurrence (see Kraemer et al., 2015 eLife). These numbers are averages based on area. However, these estimates are correlated with the values you obtain when you weight by population (cor of 0.84) so the impact on the parameter estimates will be very minor.

Specific changes to manuscript: We have clarified how the means are calculated.

Comment 3.18. *Figure S10: missing y axis label for panels A and E. Consider putting $R_{eff}=1.6$ top in the legends to follow the ordering of the curves on the plot.*

Specific changes to manuscript: We have made the suggested changes.

References

- Aledavood, Talayeh, Eduardo López, Sam G. B. Roberts, Felix Reed-Tsochas, Esteban Moro, Robin I. M. Dunbar, and Jari Saramäki. 2015. "Daily Rhythms in Mobile Telephone Communication." *PloS One* 10 (9): e0138098.
- Liu, Zhang, Ting Ma, Yunyan Du, Tao Pei, Jiawei Yi, and Hui Peng. 2018. "Mapping Hourly Dynamics of Urban Population Using Trajectories Reconstructed from Mobile Phone Records." *Transactions in GIS* 22 (2): 494–513.

REVIEWERS' COMMENTS

Reviewer #1 (Remarks to the Author):

The authors have fully addressed my comments. I do not have further questions.

Reviewer #3 (Remarks to the Author):

Very nice work, thank you for addressing my points.

If you do have a chance (assuming it's indeed accepted for publication), I wonder if you might make the blue and green colors in the box plots for figures 2 and 3 lighter? I found it really hard to see the black line in the middle of the box plots, even in the revision.

Responses to reviewers

Reviewer #1 (Remarks to the Author):

The authors have fully addressed my comments. I do not have further questions.

Reviewer #3 (Remarks to the Author):

Very nice work, thank you for addressing my points.

If you do have a chance (assuming it's indeed accepted for publication), I wonder if you might make the blue and green colors in the box plots for figures 2 and 3 lighter? I found it really hard to see the black line in the middle of the box plots, even in the revision.

Author response: We have lightened the colours.